# OsbHLH067, OsbHLH068, and OsbHLH069 redundantly regulate inflorescence axillary meristem formation in rice

**Tingting Xu**[1], **Debao Fu**[1], **Xiaohu Xiong**[1], **Junkai Zhu**[2], **Zhiyun Feng**[1], **Xiaobin Liu**[2], **Changyin Wu**[1] *

**1** National Key Laboratory of Crop Genetic Improvement and National Center of Plant Gene Research (Wuhan), Hubei Hongshan Laboratory, Huazhong Agricultural University, Wuhan, China, **2** Jiangsu Kingearth Seed Co., Ltd., Yangzhou, China

* cywu@mail.hzau.edu.cn

**Data Availability Statement:** All relevant data are within the manuscript and its Supporting Information files.

**Funding:** C. W. received the fundings from the National Natural Science Foundation of China

## Abstract

Rice axillary meristems (AMs) are essential to the formation of tillers and panicle branches in rice, and therefore play a determining role in rice yield. However, the regulation of inflorescence AM development in rice remains elusive. In this study, we identified *no spikelet 1-Dominant* (*nsp1-D*), a sparse spikelet mutant, with obvious reduction of panicle branches and spikelets. Inflorescence AM deficiency in *nsp1-D* could be ascribed to the overexpression of *OsbHLH069*. *OsbHLH069* functions redundantly with *OsbHLH067* and *OsbHLH068* in panicle AM formation. The *Osbhlh067 Osbhlh068 Osbhlh069* triple mutant had smaller panicles and fewer branches and spikelets. *OsbHLH067*, *OsbHLH068*, and *OsbHLH069* were preferentially expressed in the developing inflorescence AMs and their proteins could physically interact with LAX1. Both *nsp1-D* and *lax1* showed sparse panicles. Transcriptomic data indicated that *OsbHLH067/068/069* may be involved in the metabolic pathway during panicle AM formation. Quantitative RT-PCR results demonstrated that the expression of genes involved in meristem development and starch/sucrose metabolism was down-regulated in the triple mutant. Collectively, our study demonstrates that OsbHLH067, OsbHLH068, and OsbHLH069 have redundant functions in regulating the formation of inflorescence AMs during panicle development in rice.

## Author summary

Axillary meristems (AMs) generate branches and determine the inflorescence pattern, and further define the overall architecture of plants. In addition, they have great impacts on the tiller number and panicle size, and therefore significantly influence the seed number and yield of crops. Hence, understanding the molecular mechanism for AM development is of both scientific and application significance. Although some genes involved in panicle development of rice have been reported to date, the underlying mechanism remains largely unknown in rice. In this study, we reported that OsbHLH067, OsbHLH068, and OsbHLH069 redundantly regulate the formation of inflorescence AMs

(U20A2023, 31630054 and 31821005), the National Key Research and Development Program of Hubei Province (2022BBA54), the Natural Science Foundation of Hubei Province (2022CFA024), and the Foundation of Hubei Hongshan Laboratory (2021hszd010). The funders had no role in study design, data collection and analysis, decision to publish, or preparation of the manuscript.

**Competing interests:** The authors have declared that no competing interests exist.

in rice. *OsbHLH067*, *OsbHLH068*, and *OsbHLH069* were preferentially expressed in developing inflorescence AMs. Overexpression of *OsbHLH069* resulted in sparse panicles. The *Osbhlh067 Osbhlh068 Osbhlh069* triple mutant exhibited small panicles with fewer branches and spikelets. OsbHLH067/068/069 were found to interact with LAX1, which might be involved in the metabolism pathway and influence the gene expression related to panicle development.

## Introduction

Flowering plants can undergo reiterative growth and continuous organogenesis during their lifespan. Axillary meristems (AMs) play a central role at both the vegetative and reproductive growth stages to determine rice plant architecture. At the vegetative growth stage, AMs are initiated from the boundary between the shoot apical meristem and leaf primordium, and then develop into rice tillers. At the reproductive growth stage, after the shoot apical meristem is transformed into the inflorescence meristem (IM), inflorescence AMs hierarchically transformed into branch meristems (BMs) and spikelet meristems (SMs) and then finally develop into a rice panicle [1]. In the inflorescence architecture, the primary branch meristem initiates at the boundary between the IM and bract primordia, and then generates primary branches (PBs). Secondary branch and spikelet meristems are generated at the boundary between the elongated primary branch and bract primordia, thereby differentiating into secondary branches (SBs) and spikelets, respectively. Ultimately, the number of tillers, branches, and spikelets derived from AMs together determine the yield of rice [2].

Multiple transcriptional factors involved in inflorescence AM formation have been identified in rice. For instance, *LAX1* encodes a bHLH transcription factor and regulates AM formation during inflorescence development [3,4]. *LAX2* encodes a nuclear protein and physically interacts with LAX1 to mediate the process of AM formation [5]. *MOC1* is a transcriptional regulator of the GRAS family and mainly regulates the formation of vegetative and reproductive AMs in rice [6]. A genetic analysis has revealed that *LAX1*, *LAX2*, and *MOC1* have overlapping functions involved in distinct pathways that regulate AM formation during vegetative and reproductive development [5]. TAB1/OsWUS is another transcription factor identified for inflorescence AM formation in rice [7]. It seems that all the transcriptional factors identified in AM-defective rice mutants are conserved in various plant species. *LAX1* and *LAX2* are orthologues of the maize genes *BA1* and *BA2* respectively [8,9]; and rice *MOC1* is homologous to tomato *LS* and *Arabidopsis LAS* [6,10]. Although some of these genes are only essential for the formation of vegetative AMs but not for that of reproductive AMs, they play some conserved roles in initiating AMs in various plant species [4–6,8–11].

Auxin biosynthesis, transport, and signaling have been demonstrated to be required for inflorescence AM formation and lateral organ initiation [12]. In *Arabidopsis*, AM formation involves auxin synthesis genes *YUC1*, *4*, and *6* [13]; auxin polar transporter genes *PIN1*, and *AUX1* [14,15]; auxin polar transport regulator gene *PID* [16,17]; and the auxin signal transduction gene *MONOPTEROS* (*MP*) [18]. Notably, certain homologous genes in maize and rice also participate in inflorescence AM development. For example, mutation of the auxin biosynthesis/signaling pathway genes, including *SPI1*, *VT2*, *BIF2*, *ZMAUX1*, *BIF1*, and *BIF4*, impaired inflorescence in maize [19–23]. In rice, *OsPIN1c/d* and *OsPID* are required for AM formation during inflorescence development [24,25]. In *Arabidopsis* and maize, some transcriptional factors are associated with auxin signaling pathway to regulate inflorescence AM development. For instance, *BIF1* and *BIF4* are integral for auxin signaling modules that

dynamically regulate the expression of *BA1* [20]. However, transcriptional factors involved in the influence of phytohormone and metabolic pathway on rice AM development remain largely unknown.

The basic/helix-loop-helix (bHLH) proteins form one of the largest transcription factor families. The bHLH domain, which is composed of about 60 amino acids, enables the formation of the homodimeric or heterodimeric complex through the HLH region and determines the ability to bind downstream genes through the basic region [26]. *LAX1* encodes a bHLH protein and is a key factor determining the formation of AM in rice [3]. *LAX1* mRNA accumulates in 2–3 layers of cells in the boundary region between initiating AM and the shoot apical meristem [3,27]. LAX1 protein accumulates transiently in initiating AMs and is subsequently trafficked to the AM in a stage- and direction-specific manner for the establishment of new AMs [27]. Mutation of *LAX1* was found to severely suppress the initiation of lateral spikelets and affect both vegetative and reproductive branching [3,27]. Ectopic expression of *LAX1* also causes pleiotropic effects, including dwarfing, reduced branching, and severe sterility [3], indicating that fine regulation of *LAX1* expression is essential for normal AM formation. LAX2 is a novel nuclear protein acting synergistically with LAX1 in rice to regulate the process of AM formation [5]. SPL protein has been reported to possibly regulate *LAX1* expression directly at the transcription level [28]. Recent studies have suggested that the *LAX1* haplotype contributes to the number of panicle branches and grain weight, thereby affecting the rice yield [29,30].

In the present study, we identified a *no spikelet 1-Dominant* rice mutant (*nsp1-D*) with fewer branches and spikelets. Genetic analysis suggested that the overexpression of *OsbHLH069* resulted in the *nsp1-D* morphology. *OsbHLH069* belongs to subfamily F of the bHLH transcription factor family in rice [31], and is functionally redundant with its homologs, *OsbHLH067* and *OsbHLH068*, in regulating panicle AM formation. *In situ* hybridization results indicated that *OsbHLH067*, *OsbHLH068*, and *OsbHLH069* are preferentially expressed in the inflorescence AM, and can physically interact with LAX1 individually. In addition, the *Osbhlh067 Osbhlh068 Osbhlh069* triple mutant showed significant variations in the expression of AM formation genes such as *RCN4*, *OsSPL14*, *NL1* and *PLA1*. Our findings suggest that the OsbHLH067/068/069-LAX1 module might act through metabolism pathways such as starch and sucrose metabolism to regulate inflorescence AM development.

## Results

### Identification of a mutant with sparse panicles

A sparse panicle mutant (03Z11CH32) with few or no spikelets was identified from our T-DNA insertional mutant library [32,33]. The heterozygous mutant exhibited a semi-dominant mutation segregation with a single Mendelian locus in the progeny (sparse: normal = 210: 72, $\chi^2 = 0.08 < 3.84$ for 3:1). Therefore, the mutant was designated as *no spikelet 1-Dominant* (*nsp1-D*). Compared with WT, heterozygotes (*nsp1-D/+*) showed a weaker phenotype of sparse panicles with reduction of grain setting, and homozygous mutants *nsp1-D* displayed seriously sparse panicles (Fig 1A and 1B).

We further characterized the panicle traits of *nsp1-D* and *nsp1-D/+* plants (Fig 1C–1G). Compared with WT, these plants showed no obvious defects in the number of PBs (Fig 1C). The number of SBs was slightly reduced in *nsp1-D/+* and remarkably decreased in *nsp1-D* plants (Fig 1D). In both *nsp1-D* and *nsp1-D/+* plants, the number of spikelets in PBs (SPBs) and spikelets in SBs (SSBs) significantly decreased, ultimately resulting in a severe reduction of the number of spikelets per panicle (Fig 1E–1G). These observations suggested that defective panicle development occurred in a dose-dependent manner in *nsp1-D*.

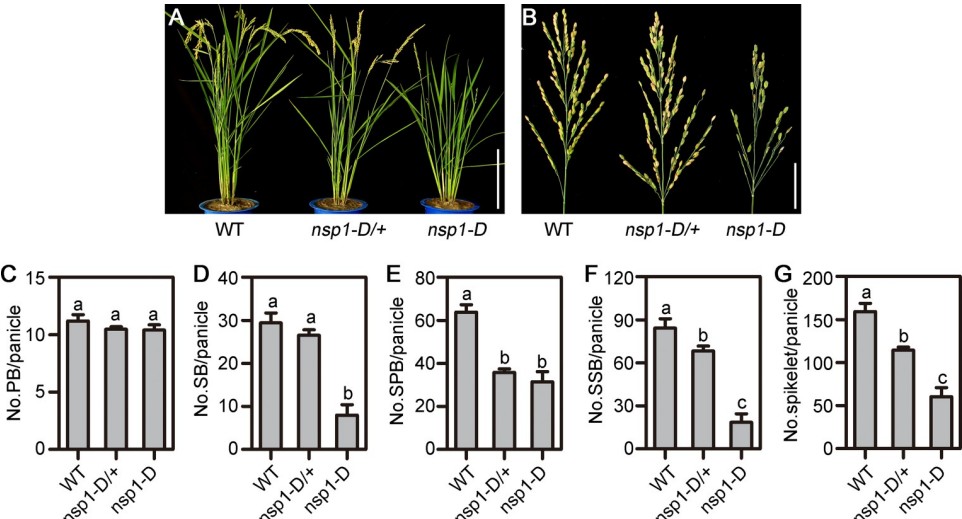

**Fig 1. Characterization of the *no spikelet 1-Dominant* (*nsp1-D*).** (A) and (B) Architecture of plant (A) and panicle (B) in wild type (WT), *nsp1-D/+*, and *nsp1-D* during reproductive growth. Bars = 20 cm in (A) and 5 cm in (B). (C) to (G) Statistical analysis of the number of primary branches (PBs) (C), secondary branches (SBs) (D), spikelets in PBs (SPBs) (E), spikelets in SBs (SSBs) (F), and total spikelets (G) per panicle among WT, *nsp1-D/+*, and *nsp1-D* plants. Values shown are the means ± SEM from 10 replicates. Letters denote significant differences as ranked by the Dunnett's test (one-way analysis of variance, P < 0.05).

## Characterization of inflorescence development in *nsp1-D*

To further explore the development of *nsp1-D* panicles, we compared the panicle development between WT and *nsp1-D* by histological section analysis and scanning electron microscopy (SEM). Histological section analysis revealed no remarkable morphological differences in apices between WT and *nsp1-D* (Fig 2A and 2E). At the reproductive stage, WT showed normal generation of PBs (Fig 2B), but PB primordia in *nsp1-D* were largely suppressed (Fig 2F). In addition, the SB primordia (Fig 2G) and spikelet primordia (Fig 2H) were significantly reduced in *nsp1-D* compared with in WT (Fig 2C and 2D). SEM results further demonstrated that nearly no SB primordia were formed, and there were only elongated PB primordia in *nsp1-D* at the spikelet primordia formation stage (Fig 2I–2P). These results suggested that inflorescence AM formation was compromised in *nsp1-D*.

*OSH1* is considered as a marker gene of meristematic cells in rice [34]. We therefore examined the expression pattern of *OSH1* in *nsp1-D* by *in situ* hybridization. *OSH1* signals could be detected in PB and SB meristems in WT plants (Fig 2Q and 2R). However, in *nsp1-D*, these signals were greatly reduced in PB meristem, and even not detected in the undeveloped meristem of SB and spikelets (Fig 2S and 2T). Taken together, it could be speculated that the corresponding gene in *nsp1-D* may be involved in the inflorescence AM formation in rice.

## Identification of the *NSP1* gene

Thermal asymmetric interlaced PCR (Tail-PCR) was performed to identify the T-DNA insertion site in *nsp1-D* [33]. The flanking sequence of the T-DNA insertion site indicated the presence of a truncated T-DNA at –6138 bp in the promoter of the LOC_Os01g57580 gene in *nsp1-D* (Fig 3A). PCR amplification results suggested that the insertion was well co-segregated with the panicle morphology in the progenies (n = 20) of the *nsp1-D/+* plant (Fig 3B). All the T-DNA insertion homozygotes showed severely sparse panicles, and heterozygotes exhibited a weaker phenotype of sparse panicles. Quantitative RT-PCR (qRT-PCR) analysis revealed a notable increase in the

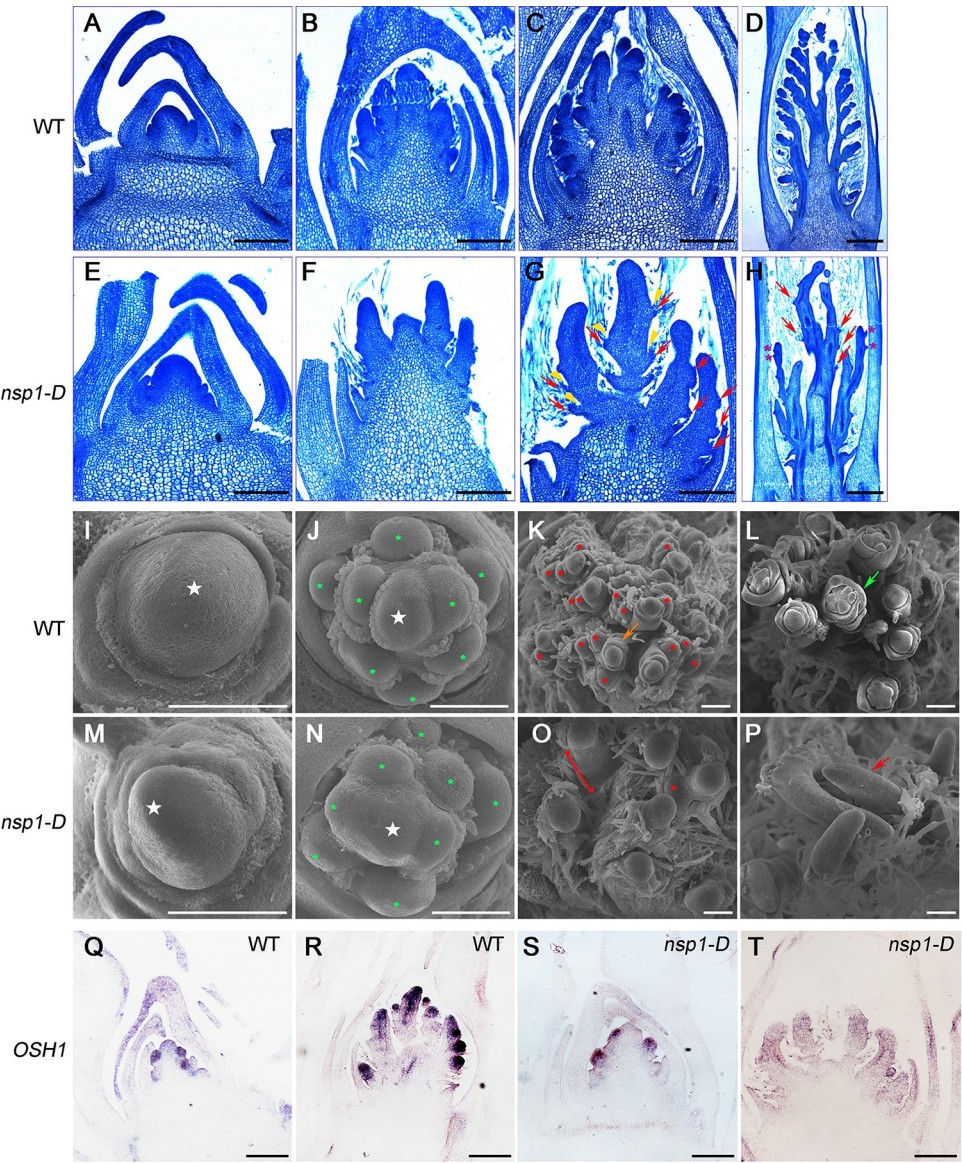

**Fig 2. Morphological analysis of the *nsp1-D*.** (A) to (H) Longitudinal section of a developing inflorescence in WT (A to D) and *nsp1-D* plants (E to H) during formation of the first bract primordium (A, E), the primary branch primordia (B, F), the secondary branch primordia or spikelet primordia (C, G), and the floret primordia (D, H). Red arrows (in G, H) indicate the empty bracts; Yellow arrowheads (in G) indicate the positions where the axillary meristems should initiate. Purple stars (in H) indicate the bract-like knobs. Bars = 200 μm. (I) to (P) Scanning electron microscopy images showing the inflorescence development for WT (I to L) and *nsp1-D* plants (M to P) during formation of the first bract primordium (I, M), the primary branch primordia (J, N), the secondary branch primordia or spikelet primordia (K, O), and the floret primordia (L, P). White stars indicate inflorescence meristem; green and red asterisks denote primary and secondary branch primordia, respectively; the red bracket outlines the region where secondary branch primordia failed to initiate; the yellow, green, and red arrows denote spikelet primordia, floret primordia, and elongated primary branch primordia, respectively. Bars = 100 μm. (Q) to (T) *In situ* localization of *OSH1* in WT (Q, R) and *nsp1-D* (S, T) inflorescences at the stage of primary branch meristem (Q, S) and secondary branch meristem (R, T) differentiation. Bars = 100 μm.

expression of LOC_Os01g57580, and normal expression of other genes surrounding the T-DNA insertion site in the 100 kb region of *nsp1-D* relative to that in WT (Fig 3C). Therefore, LOC_Os01g57580 might be the gene responsible for the sparse panicle phenotype of *nsp1-D*.

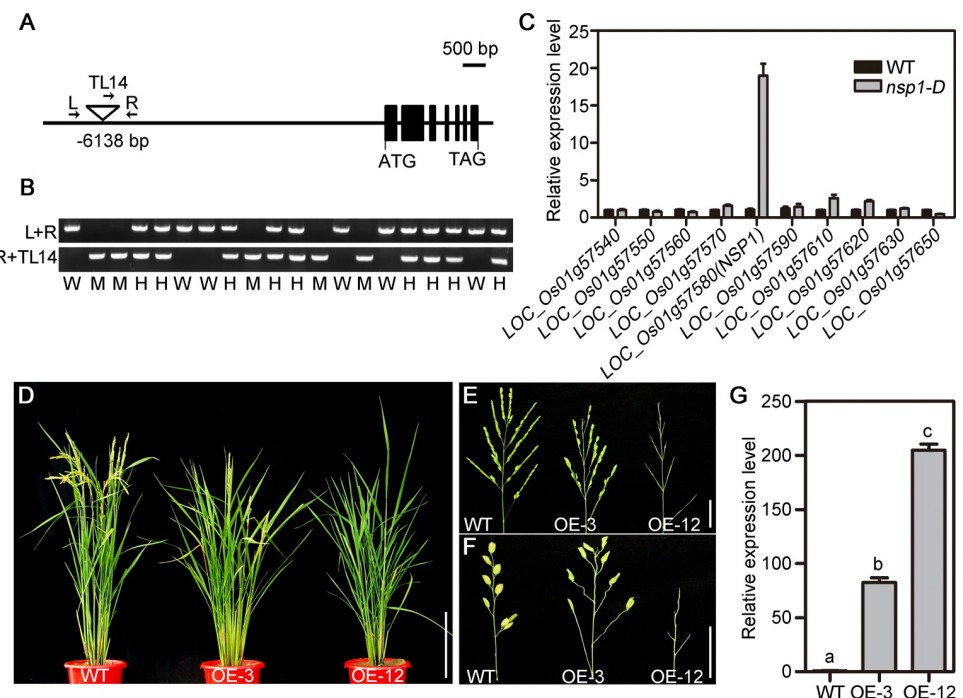

**Fig 3. Identification of *No Spikelet 1* (*NSP1*).** (A) Structure of the *NSP1* genome and the T-DNA insertion site. Black boxes represent exons; lines between the boxes represent introns; the inverted triangle indicates T-DNA. Primers L and R on the *NSP1* genome and primer TL14 at the T-DNA left border used for genotype analysis are marked with arrows. (B) Co-segregation analysis of *nsp1-D/+*. W, H, and M indicate wild type (WT), heterozygous, and homozygous for T-DNA insertion, respectively. (C) Quantitative RT-PCR analysis of genes flanking the T-DNA insertion site in young panicles (< 5 mm) of WT and *nsp1-D*. The internal rice *Ubiquitin* (*UBQ*) gene was used to normalize gene expression. Data are the means ± SEM from nine replicates. (D) to (F) Plant morphology (D) and panicle morphology (E) of *35S-pOsbHLH069::OsbHLH069* transgenic plants (OE-3 and OE-12). (F) Closeup view of one primary branch in (E). (G) Expression analysis of *OsbHLH069* in the leaves of the *35S-pOsbHLH069::OsbHLH069* transgenic plants (OE-3 and OE-12). Gene expression was normalized to the rice *UBQ* gene. Values shown indicate the means ± SEM from three replicates. Different letters denote significant differences ranked by the Dunnett's test (one-way analysis of variance, P < 0.05). Bars = 20 cm in (D) and 4 cm in (E, F).

LOC_Os01g57580 encodes a typical bHLH transcription factor, and is designated as *OsbHLH069* [31]. We then overexpressed *OsbHLH069* driven by the CaMV35S (*35S-pOsbHLH069::OsbHLH069*) in rice. Among the 45 putative transgenic plants, 13 positive transgenic plants exhibited obvious sparse panicle phenotype. We selected two independent transgenic lines (OE-3 and OE-12, heterozygous and homozygous of *35S-pOsbHLH069::OsbHLH069*, respectively) for further examination. Compared with negative transgenic plants, the OE-3 plant showed a mild phenotype with a few branches and spikelets, and the OE-12 plant displayed a severe phenotype without SBs and spikelets (Fig 3D–3F). qRT-PCR analysis demonstrated that OE-3 and OE-12 plants had significantly higher *OsbHLH069* transcript levels than WT plants (Fig 3G). Based on these results, it could be concluded that *OsbHLH069* is *NSP1*, and its overexpression would result in sparse panicles in *nsp1-D* plants.

## *OsbHLH067*, *OsbHLH068*, and *OsbHLH069* are preferentially expressed in inflorescence AM

It has been reported that the rice genome contains 167 bHLH genes, which can be subdivided into 22 subfamilies named as A–V [31]. OsbHLH069 belongs to subfamily F (14 bHLH

proteins), and OsbHLH067, OsbHLH068, and OsbHLH070 are closely homologous to OsbHLH069 (S1A Fig).

Alignment analysis revealed that OsbHLH067, OsbHLH068, and OsbHLH069 contain a conserved bHLH domain (S1B Fig), but OsbHLH070 contains an atypical bHLH domain that does not bind DNA (S1C Fig) [26]. qRT-PCR results demonstrated that *OsbHLH067*, *OsbHLH068*, and *OsbHLH069* are constitutively expressed in various organs, including the root, culm, leaf, leaf sheath, and young panicles of less than 5 mm (S2A–S2C Fig). We subsequently carried out *in situ* hybridization for more precise examination of the spatial expression patterns of *OsbHLH067*, *OsbHLH068*, and *OsbHLH069* in young panicles. Just like *OSH1*, *OsbHLH069* (Fig 4A–4D) and *OsbHLH067* (Fig 4E–4H) were expressed in all reproductive meristems, such as IM, PB meristem, SB meristem, and SM (Fig 4M–4P) [3]. *OsbHLH068* mRNA showed similar expression patterns to *OsbHLH069* and *OsbHLH067* except for in the IM (Fig 4I–4L). Moreover, *LAX1* mRNA was preferentially accumulated in the boundary of initiating AM (Fig 4Q–4T) [3]. Movement of the LAX1 protein towards the future AM has been reported to be required for maintaining the AM development [27]. Our results suggested that *OsbHLH067/068/069* are preferentially expressed in inflorescence AM, and may act together with LAX1 for the AM development.

LAX1 has been reported to be predominantly localized in the nucleus [27]. We fused LAX1 with RFP as a nuclear marker, and then investigated the subcellular localization of OsbHLH067, OsbHLH068, and OsbHLH069 by individually expressing their fusion proteins with GFP driven by a CaMV35S promoter in rice protoplasts. Compared with the empty GFP protein evenly distributed in the cytoplasm and nucleus, the GFP-fused proteins were co-localized with RFP-fused LAX1 in the nucleus (S2D Fig), suggesting that OsbHLH067, OsbHLH068, and OsbHLH069 may function in the nucleus.

### *OsbHLH067/068/069* redundantly regulate inflorescence AM formation

Considering that *OsbHLH067*, *OsbHLH068*, and *OsbHLH069* are homologous genes with similar expression patterns during inflorescence development in rice, we speculated that they might be involved in panicle AM development. Hence, CRISPR/Cas9 system was used to generate single, double, and triple mutants for them (S3 Fig). Compared with WT plants, the single and double mutants showed no noticeable change in morphology (Fig 5A–5H, S4 Fig). However, the triple mutant displayed severe defects, including dwarf stature, single culm, and small panicle with few branches and spikelets (Fig 5A–5H). Histological analysis revealed that the triple mutant showed significant reduction of branch primordia, indicating that inflorescence AM formation is compromised in the absence of *OsbHLH067*, *OsbHLH068*, and *OsbHLH069* (Fig 5I). In addition, *in situ* hybridization of *OSH1* mRNA suggested that AM formation was arrested in the panicles of the triple mutant (Fig 5J). Taken together, our results suggest that *OsbHLH067*, *OsbHLH068*, and *OsbHLH069* are functionally redundant for inflorescence AM formation.

### Genetic interaction between *OsbHLH069* and *LAX1*

*LAX1* acts as a major regulator on AM formation in rice [3,27]. In our T-DNA insertion mutant library, an identified allelic mutant of *lax1* showed reduction of branches and spikelets (S5A–S5G Fig). To examine the genetic interaction between *OsbHLH069* and *LAX1*, we attempted to generate a double mutant of *nsp1-D/+* and *lax1* (Fig 6). Given the close physical locations of *OsbHLH069* and *LAX1* on chromosome 1, we failed to obtain the *nsp1-D lax1* double mutant. As described above, compared with WT plants, *nsp1-D* and *nsp1-D/+* showed reduction of SPBs and SSBs in panicle (Figs 1A–1G and 6A–6I). The defects in lateral

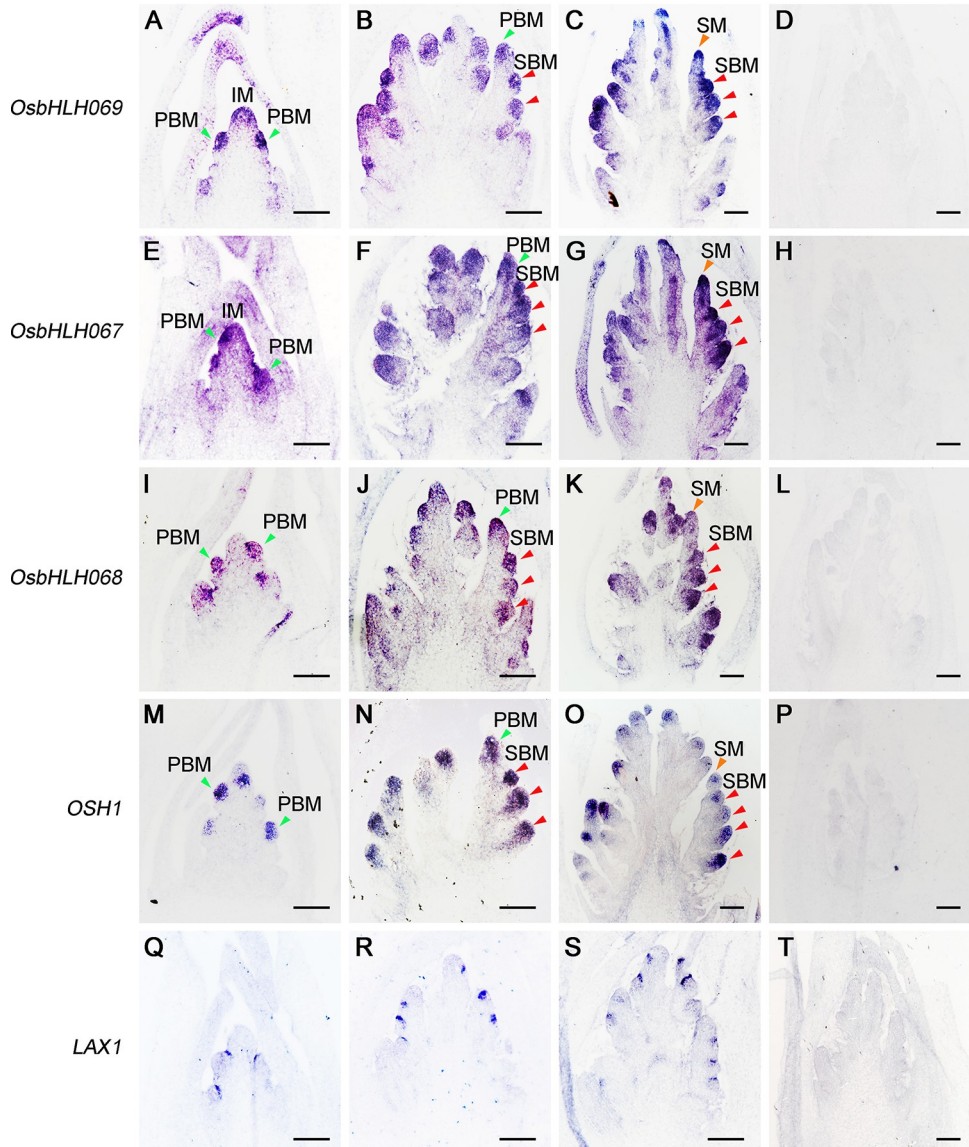

**Fig 4. Spatial expression patterns of *OsbHLH067*, *OsbHLH068*, and *OsbHLH069* in young panicles.** *In situ* localization of *OsbHLH069* (A to D), *OsbHLH067* (E to H), *OsbHLH068* (I to L), *OSH1* (M to P), and *LAX1* (Q to T) in developing inflorescences. (D), (H), (L), (P) and (T) The sense probes of *OsbHLH069* (D), *OsbHLH067* (H), *OsbHLH068* (L), *OSH1* (P), and *LAX1* (T) served as controls. (A), (E), (I), (M) and (Q) A developing inflorescence at the primary branch meristem (PBM, green triangle) differentiation stage. (B), (F), (J), (N) and (R) A developing inflorescence at the secondary branch meristem (SBM, red triangle) differentiation stage. (C), (G), (K), (O) and (S) A developing inflorescence at the spikelet meristem (SM, yellow triangle) initiation stage. IM, inflorescence meristem. Bars = 100 μm.

branching of the panicle became more severe when *lax1* or *lax1/+* was combined with *nsp1-D* and *nsp1-D/+* (Fig 6), indicating that *OsbHLH069* and *LAX* might have a synergistic effect on panicle AM formation.

We then investigated the possibility of transcriptional regulation among *OsbHLH067*, *OsbHLH068*, *OsbHLH069*, and *LAX1*. qRT-PCR analysis showed that the expression of *OsbHLH067*, *OsbHLH068*, and *OsbHLH069* was not affected in *lax1* plants (S5H Fig). In the *Osbhlh067 Osbhlh068 Osbhlh069* triple mutant plants, the transcription level of *LAX1* was

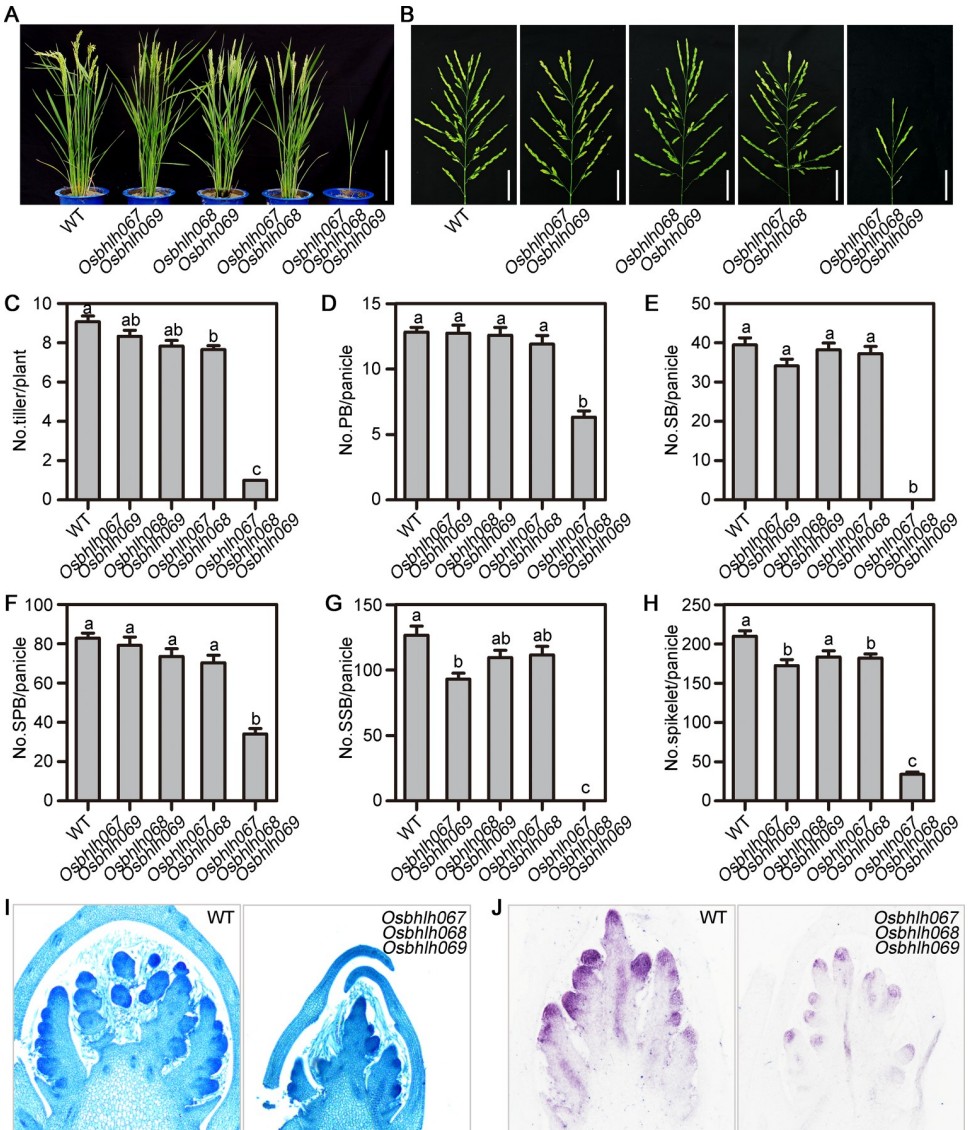

**Fig 5. Characterization of double and triple mutants of *OsbHLH067*, *OsbHLH068*, and *OsbHLH069*.** (A) and (B) Phenotype comparisons of the plant (A) and panicle (B) for wild type (WT), *Osbhlh067 Osbhlh069*, *Osbhlh068 Osbhlh069*, and *Osbhlh067 Osbhlh068* double mutants, and the *Osbhlh067 Osbhlh068 Osbhlh069* triple mutant. Bars = 20 cm in (A) and 4 cm in (B). (C) to (H) Quantification of the number of (C) tillers, (D) primary branches (PBs), (E) secondary branches (SBs), (F) spikelets in PBs (SPBs), (G) spikelets in SBs (SSBs) and (H) total spikelets per panicle in WT, *Osbhlh067 Osbhlh069*, *Osbhlh068 Osbhlh069*, and *Osbhlh067 Osbhlh068* double mutants, and the *Osbhlh067 Osbhlh068 Osbhlh069* triple mutant. Data are the means ± SEM from 12 replicates. Different letters denote significant differences ranked by the Dunnett's test (one-way analysis of variance, $P < 0.05$). (I) Paraffin sections of inflorescences with SB primordia from WT and the *Osbhlh067 Osbhlh068 Osbhlh069* triple mutant. Bars = 100 μm. (J) *In situ* localization of *OSH1* in WT and *Osbhlh067 Osbhlh068 Osbhlh069* triple mutant panicles at the secondary branch primordia differentiation stage. Bars = 100 μm.

slightly increased as indicated by qRT-PCR (S5I Fig). *In situ* hybridization analysis revealed that the accumulation of *LAX1* mRNA was not affected in the few AMs of the triple mutant (S5J–S5L Fig). These results indicated that *OsbHLH067/068/069* and *LAX1* have no significant mutual effect at the transcriptional level.

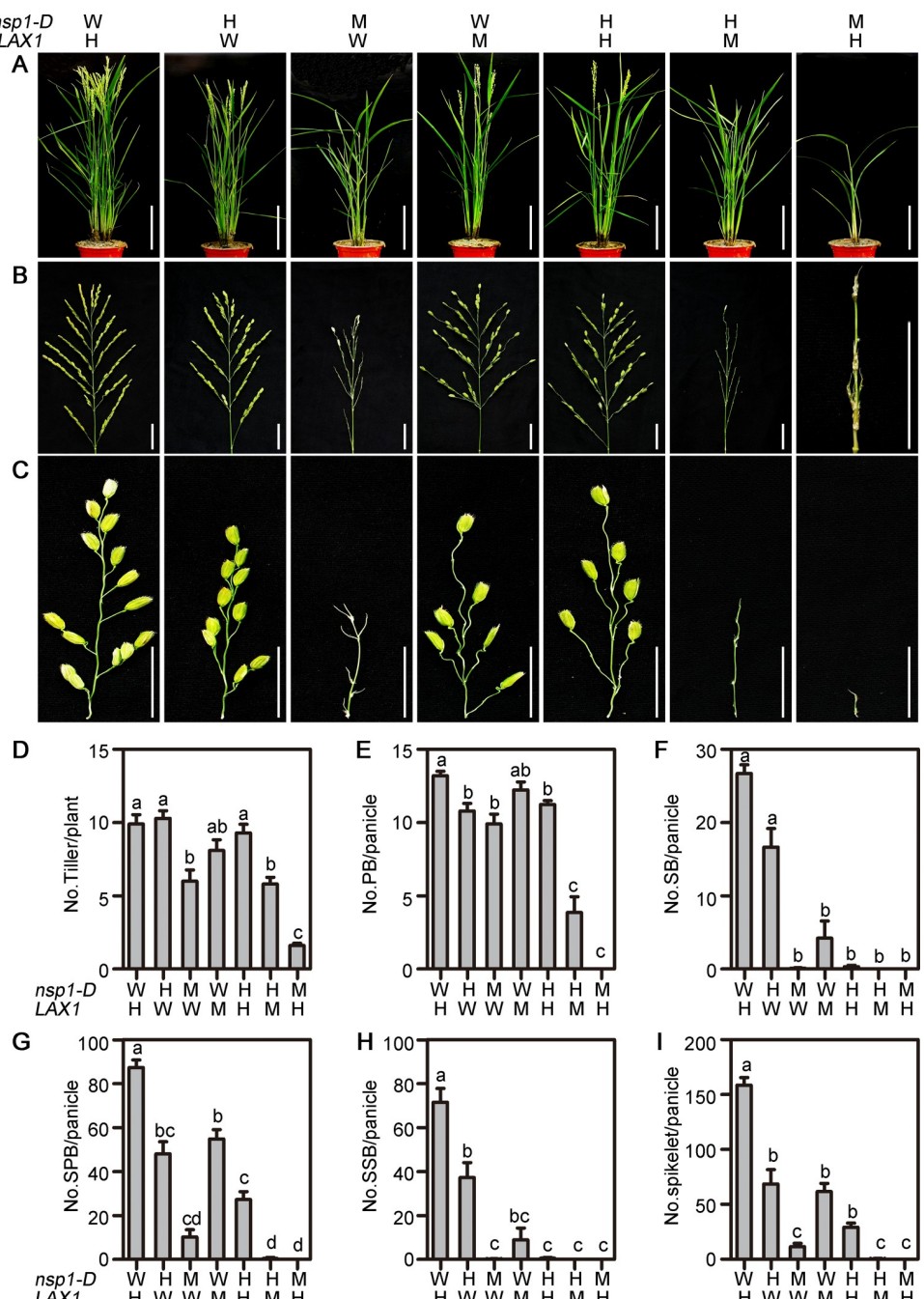

**Fig 6. Phenotype analysis of plants in the progenies of *nsp1-D/+ lax1/+*.** (A) to (C) Architectures of (A) the plant, (B) panicle, and (C) primary branch (PB) in the progeny. (D) to (I) Quantitative statistics of the number of (D) tillers, (E) PBs, (F) secondary branches (SBs), (G) spikelets of PBs (SPBs), (H) spikelets of SBs (SSBs) and (I) total spikelets in the progeny. Data are the means ± SEM from 10 replicates. Different letters denote significant differences ranked by the Dunnett's test (one-way analysis of variance, P < 0.05). WH, *NSP1 lax1/+*; HW, *nsp1-D/+ LAX1*; MW, *nsp1-D LAX1*; WM, *NSP1 lax1*; HH, *nsp1-D/+ lax1/+*; HM, *nsp1-D/+ lax1*; MH, *nsp1-D lax1/+*. Bars = 20 cm in (A) and 3 cm in (B, C).

## OsbHLH067/068/069 physically interact with LAX1

Considering that LAX1 moves directionally to the newly formed AM [27], where it might have overlapping functions with OsbHLH067/068/069, it is possible that OsbHLH067/068/069 physically interact with LAX1 in the developing AM. To test this hypothesis, we carried out a yeast two-hybrid assay. Because OsbHLH067, OsbHLH068, OsbHLH069, and LAX1 all showed self-activation activities in yeast cells (Fig 7A), we co-transformed a truncated LAX1 fragment (1–159 aa) without self-activation activity as a bait in yeast cell, with the preys OsbHLH067/068/069 (Fig 7B). The resultant transformants all grew on the diluted selective medium, except for the control transformant of pGAD-T7 or pGAD-OsbHLH003 (Fig 7B), suggesting that OsbHLH067/068/069 interact with LAX1 in yeast cells. We further used the truncated LAX1 (31–99 aa) containing the bHLH domain (41–89 aa) as a bait for the interaction assay. The results showed that OsbHLH067/068/069 could interact with the bHLH domain of LAX1, but not with the control of pGAD-OsbHLH003 (Fig 7B).

We then carried out bimolecular fluorescence complementation (BiFC) assays to confirm the interaction between OsbHLH067/068/069 and LAX1 in rice protoplasts. As anticipated, reconstructed CFP fluorescence was detected in the nuclei of rice protoplasts when co-transforming cCFP-LAX1 with OsbHLH067-nCFP, OsbHLH068-nCFP, or OsbHLH069-nCFP (Fig 7C). We further confirmed the interaction between OsbHLH69/067/068 and LAX1 by pull down assays, respectively (Fig 7D and 7E). Taken together, our results suggested that OsbHLH067/068/069 physically interact with LAX1 and might act synergistically with it to regulate panicle AM formation in rice.

## The OsbHLH067/068/069-LAX1 module is associated with starch and sucrose metabolism

To further understand the biological functions of OsbHLH067/068/069 required for AM development, we used young panicles ($\leq$ 2 mm) from the *Osbhlh067 Osbhlh068 Osbhlh069* triple mutant and WT plants to perform a comparative RNA-seq. Principal component (PC) analysis demonstrated that the RNA-seq data from three biological replicates were closely clustered (Fig 8A). Compared with WT, the triple mutant was found to contain 1533 down-regulated and 1526 up-regulated genes (Q value $\leq$ 0.05, fold change > 1.5; Fig 8B, S1 Table). Kyoto Encyclopedia of Genes and Genomes (KEGG) enrichment results suggested that these down-regulated genes were significantly enriched in multiple biological processes, including secondary metabolite biosynthesis, fatty acid metabolism, alanine aspartate and glutamate metabolism, starch and sucrose metabolism, cell cycle, and phototransduction (Q value $\leq$ 0.05; Fig 8C, S2 Table). Up-regulated genes were significantly enriched only in "protein processing in endoplasmic reticulum" (Q value $\leq$ 0.05) (Fig 8D, S3 Table).

Because OsbHLH067, OsbHLH068, and OsbHLH069 are typical transcription factors with transcriptional activation activities (Fig 7A), we examined these down-regulated genes in the triple mutant by a comparison with WT by qRT-PCR analysis (S4 Table). As expected, the expression levels of the selected genes (sorted by Q value) were confirmed to be down-regulated in the triple mutant (Fig 8E and 8F). Interestingly, all the examined genes involved in sucrose metabolism were significantly down-regulated in *lax1* (Fig 8F). For the selected genes enriched in the other biological processes, they showed no obvious change in expression in *lax1* compared with WT (Fig 8E). Among the down-regulated genes in the triple mutant, some of them have been suggested to play important roles in regulating panicle development, such as *RCN4*, *OsTB1*, *OsSPL14*, *NL1*, and *PLA1* [35–42]. Their expression levels were also significantly reduced in both the triple mutant and *lax1* compared with WT (Fig 8F, S1 Table). Thus, we speculate OsbHLH067/068/069 function redundantly and might be involved in

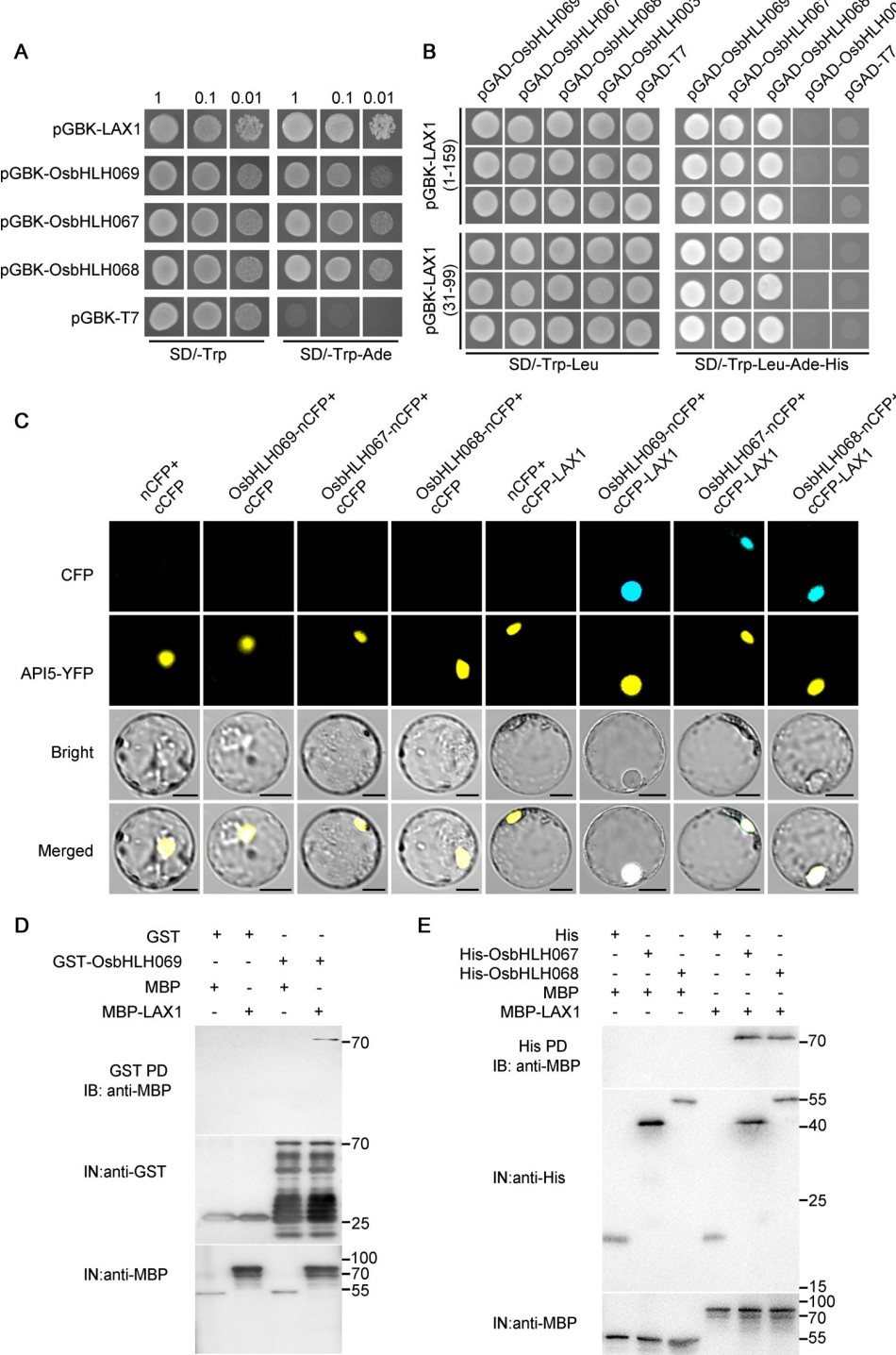

**Fig 7. Molecular interactions between OsbHLH067/068/069 and LAX1.** (A) Transcription activity assay of OsbHLH067, OsbHLH068, OsbHLH069, and LAX1 in yeast cells. The coding sequences (CDS) of *LAX1*, *OsbHLH067*, *OsbHLH068*, and *OsbHLH069* were introduced into the pGBK-T7 vector, respectively. The empty vector pGBK-T7 served as a negative control. Both vectors were transformed into AH109, respectively. Cultures were diluted (1:10 successive dilution series) and spotted onto the control medium without Trp and selective medium without Trp and Ade. (B) In yeast two-hybrid assays, OsbHLH067/068/069 interacts with LAX1 through the bHLH domain of LAX1. The pGBK-LAX1(1–159)/pGBK-LAX1(31–99) and pGAD-OsbHLH003 combinations were separately used as negative controls. (C) Bimolecular fluorescence complementation (BiFC) analysis shows the interaction between OsbHLH067/068/069 and LAX1 in rice protoplasts. API5-YFP served as a nuclear marker. Bars = 10 μm. (D) and (E)

In vitro pull-down assay showing the direct interaction between OsbHLH067/068/069 and LAX1. MBP-LAX1 was pulled down (PD) by GST-OsbHLH069 immobilized on glutathione-conjugated agarose beads and by His-OsbHLH067/068 immobilized on Ni sepharose beads, and analyzed by immunoblotting (IB) using an anti-MBP antibody. Each input (IN) lane was immunoblotted using an anti-His, anti-GST, or anti-MBP antibody.

starch and sucrose metabolism process to modulate AM development by interacting with LAX1.

## Discussion

### The function of *OsbHLH067*, *OsbHLH068*, and *OsbHLH069* in plants

The bHLH protein transcription factors are ubiquitous transcriptional regulators that control many different developmental and physiological processes [43], which can be divided into 32 subfamilies [44]. Usually, the *bHLH* genes in the same subfamily participate in the same biological process with partial or complete functional redundancy. For instance, all six members in the bHLH subfamily 16 have the conserved function of regulating flag leaf angle in rice [45]. *Arabidopsis* PIF-family members PIF1, PIF3, PIF4, and PIF5 (belonging to subfamily 24) regulate seedling morphogenesis through differential expression-patterning of shared target genes [46]. The *bHLH010*, *bHLH089*, and *bHLH091* genes from bHLH subfamily 9 are redundantly required for *Arabidopsis* anther development [47]. In our study, *OsbHLH067*, *OsbHLH068*, *OsbHLH069*, and *OsbHLH070* belong to the subfamily F of the bHLH transcription factor in rice, although *OsbHLH070* does not contain a typical bHLH domain [26]. *OsbHLH069* was demonstrated to function redundantly with *OsbHLH067* and *OsbHLH068*, which are all involved in panicle AM development. Because we have not obtained the *Osbhlh070*, the role of *OsbHLH070* in inflorescence AM development cannot be ruled out.

Notably, rice bHLH subfamily F and *Arabidopsis* bHLH subfamily X [48] are clustered into the phylogenetic clade 15 [44]. A recent study reported that some members of subfamily X in *Arabidopsis* determine the competence of the pericycle for lateral root initiation [49]. Overexpression of *AtbHLH112* suppressed lateral root emergence, but the *Atbhlh112* mutant exhibited no obvious defect [50]. OsbHLH067, OsbHLH068, and OsbHLH069 are the close homologs to *Arabidopsis* AtbHLH112 [31]. *OsbHLH068* and *AtbHLH112* showed a similar expression patterns in transgenic *Arabidopsis* and partially redundant functions in salt stress response and opposite functions in flowering transition [51]. In this study, overexpression of *OsbHLH069* resulted in an obvious defect in AM development (Figs 1, 2 and 3) and partially delayed flowering (Fig 3D). These observations can be explained by the fact that genes in the bHLH subfamily potentially have redundant but distinct functions in rice and *Arabidopsis*, presumably due to the evolutionary functional divergence of homolog-encoded proteins.

### OsbHLH067/068/069 may participate in meristem development through metabolism-related pathways

*In situ* hybridization revealed that *OsbHLH067/068/069* was predominantly expressed in both inflorescence AM and IM (Fig 4). We suspect that OsbHLH067/068/069 may be mainly involved in inflorescence meristem development. RNA-Seq data analysis demonstrated that the differentially expressed genes in the *Osbhlh067 Osbhlh068 Osbhlh069* triple mutant were significantly enriched in multiple metabolic pathways such as amino acid metabolism, fatty acid metabolism, secondary metabolism, and cell cycle (Fig 8C). qRT-PCR analysis further confirmed the downregulation of those genes involved in starch and sucrose metabolic pathway in the triple mutant (Fig 8F). Although the functions of genes associated with AM

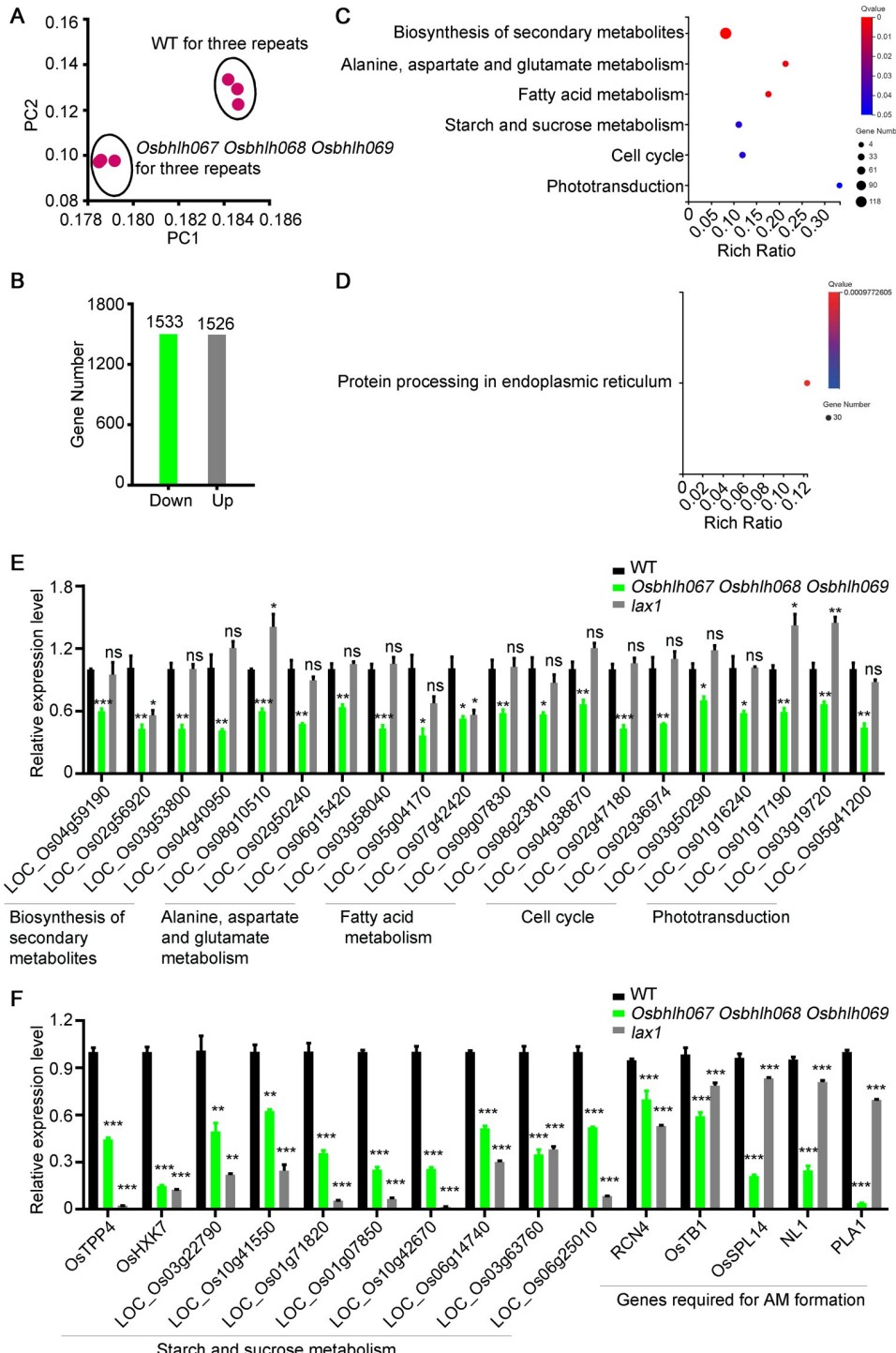

**Fig 8. Transcriptome analyses for the *Osbhlh067 Osbhlh068 Osbhlh069* triple mutant panicles.** (A) Principal component analysis of transcriptome data with three biological replicates in the young panicles (< 2 mm) of *Osbhlh067 Osbhlh068 Osbhlh069* triple mutant. Samples with high similarity will be clustered together. (B) Number of DEGs (Q value ≤ 0.05 and fold change >1.5) between WT and *Osbhlh067 Osbhlh068 Osbhlh069* triple mutant. (C) and (D) Representative KEGG terms (Q value ≤ 0.05) from the down-regulated genes (C) and the up-regulated genes (D) in the *Osbhlh067 Osbhlh068 Osbhlh069* triple mutant. (E) Quantitative RT-PCR analysis of genes involved in secondary metabolite biosynthesis, alanine aspartate and glutamate metabolism, fatty acid metabolism, cell cycle, and phototransduction in the *Osbhlh067 Osbhlh068 Osbhlh069* triple mutant and *lax1* panicles. (F) Quantitative RT-PCR analysis of genes involved in meristem formation and starch and sucrose metabolism for WT and the *Osbhlh067*

*Osbhlh068 Osbhlh069* triple mutant. Data in (E) and (F) were normalized to the rice *UBQ* gene, and values represent the means ± SEM from three replicates. Significant difference (two-tailed Student's *t*-test, *$P < 0.05$, **$P < 0.01$, ***$P < 0.001$); ns indicates not significant (two-tailed Student's *t*-test; $P > 0.05$).

development have not been reported in rice, some of their homologs involved in meristem development in other plants have been characterized. The maize genome encodes 14 trehalose-6-phosphate synthase (TPS) genes and 11 trehalose-6-phosphate phosphatase (TPP) genes. *ZmTPP4* is the ortholog of *OsTPP4* in maize [52]; *ZmTPP4* is a complete paralogue of *RA3*; and loss of *ZmTPP4* and *RA3* would reduce meristem determinacy and increase inflorescence branching [53,54]. The level of trehalose-6-phosphate levels influences plant growth and development through perturbations of the glucose sensor HEXOKINASE 1 (HXK1). *HXK1* over-expressing lines exhibited an increase in the number of primary rosette branches despite of no elevation of sugar levels in *Arabidopsis* [55,56]. In this study, the expression of *OsTPP4* and *OsHXK7* was significantly suppressed in both the triple mutant and *lax1*. Multiple investigations have demonstrated that bud outgrowth might be mediated by specific nutritional and hormonal signaling pathways [55]. Thus, we conjecture that OsbHLH067/068/069 influence AM development partially through the starch and sucrose metabolism pathway.

## Interactions between OsbHLH067/068/069 and LAX1 in regulating inflorescence AM development

In rice, AM development during the reproductive stage entails stem cell maintenance, AM initiation, AM differentiation, and branching outgrowth. Previous studies have demonstrated that *LAX1* mRNA is restricted within a few layers of cells at the boundary region between the SAM and initiating AM [3] and that LAX1 protein accumulates transiently in initiating AM [27]. Here, we found that *LAX1* mRNA indeed accumulates in the boundary region where AM occurs (Fig 4Q–4S). Considering the movement of LAX1 protein, OsbHLH067/068/069 are proposed to interact with LAX1 protein, which coordinates AM development.

This study investigated the interaction between *OsbHLH067/068/069* and *LAX1* for AM formation. Like the mutation of *LAX1* gene, the *Osbhlh067 Osbhlh068 Osbhlh069* triple mutant plants showed significant decreases in SBs rather than lateral spikelets on PBs (Fig 5) [3]. Thus, it can be deduced that the interaction of OsbHLH067/068/069 and LAX1 may have certain spatiotemporal characteristic, and determines their transcriptional activity or their ability to bind specific target genes. Indeed, our RNA-seq data and qRT-PCR analysis suggested that five unigenes annotated as contributors to inflorescence architecture in plants were down-regulated in the triple mutant and *lax1* (S1 Table, Fig 8F). Among them, *RCN4*, a TFL1-like gene, acts as a direct downstream target of OsMADS5 and OsMADS34 and precisely regulates inflorescence and SM determinacy [42]. *OsTB1* was found to work downstream strigolactones to inhibit the outgrowth of axillary buds and increase the inflorescence size and spikelet number in rice [36,41]. An appropriate increase in the expression of *OsSPL14* at the reproductive stage could promote panicle branching and higher grain yield in rice [35,37]. *PLASTOCHRON1* (*PLA1*), a cytochrome P450 gene, is mainly expressed in bracts of the panicle and acts as a timekeeper of panicle development [38]. A recent study has shown that bract suppression regulated by the miR156/529-SPLs-NL1-PLA1 module is required for the transition from vegetative to reproductive branching in rice [40]. In addition, it has been reported that both *OsTB1* and *OsSPL14* control panicle development through strigolactone signaling and sugar sensing [39]. RNA-seq data and qRT-PCR results demonstrated that some genes involved in the starch and sucrose metabolic pathway were also downregulated in the triple mutant and *lax1*

(Fig 8F). Therefore, we speculate that the interaction of OsbHLH067/068/069 with LAX1 is required to regulate AM development.

## Temporal-spatial expression of OsbHLH067/068/069 is essential for regulating panicle branching

It has been demonstrated that inflorescence AM formation are mediated by phytohormones [57]. In maize, BIF1 and BIF4, two AUXIN/INDOLE-3-ACETIC ACID (Aux/IAA) proteins, are involved in the regulation of the BA1 orthologous of *LAX1*, suggesting that auxin signaling modules are directly responsible for AM formation [20]. Loss-of-function or overexpression of *MONOPTEROS (MP)/ARF5* led to strongly suppressed reproductive AM initiation, resulting in "pin"-like inflorescence in *Arabidopsis* [18,58]. These findings suggest that auxin plays an essential role in ensuring the proper inflorescence architecture of plants. Considering that rice *LAX1* is the ortholog of *BA1*, we hypothesized that *LAX1* acts a conserved mechanism in boundary domains for AM formation by the auxin pathway. qRT-PCR analysis suggested some auxin-related genes were consistently suppressed in *lax1* plants (S6A Fig). In contrast, most of them have normal expression levels in the triple mutant *Osbhlh067 Osbhlh068 Osbhlh069* triple mutant (S6B Fig). These results indicated that *LAX1* might participate in auxin pathway alone, but not through the interaction with *OsbHLH067/068/069*.

The inflorescence architecture is determined by meristem size, bud initiation and outgrowth, and controlled by endogenous and external factors. After transition from vegetative to reproductive development, rice panicle development is set to undergo an interned process, that is IM initiates indeterminate BMs, which in turn produce a series of SMs [57]. Therefore, the comprehensive genetic networks associated with inflorescence architecture must be precisely regulated. In rice, *LAX1* might be directly regulated by *SPL14* [59]. Both RNAi and overexpression lines of SPL genes showed remarkably reduced panicle branches, indicating that the expression of *LAX1* must be fine-tuned for reproductive branching [59]. Genetic analysis has revealed that mutation in *LAX1* severely suppresses the initiation of lateral spikelets and affects both vegetative and reproductive branching [3,4]. Overexpression of *LAX1* also causes reduction of branching [3]. Our results suggest that OsbHLH067/068/069 physically interact with LAX1 and may tightly regulate LAX1 activity to control panicle AM formation (Fig 7B–7E). Both the *OsbHLH069*-overexpressing plants and the *Osbhlh067 Osbhlh068 Osbhlh069* triple mutant showed some defects in reproductive AM development, resulting in reduction of panicle branches and spikelets (Figs 1, 3 and 5). OsbHLH067/068/069 have broad expression pattern including IM, PB meristem, SB meristem, and SM (Fig 4A–4P), which suggest that they may have multiple functions in regulating reproductive branching. *FRIZY PANICLE* (*FZP*) is required to establish SM by inhibiting the formation of BMs [60]. *TAWAWA1* is a unique regulator of SM phase transition in rice [61]. *OsMADS1* is first expressed in SMs and required for floral meristem identity [62]. To elucidate the function of OsbHLH067/068/069 on SM development, the expression patterns of *TAWAWA1*, *OsMADS1* and *FZP*, would be examined in the future.

In this study, we examined that the OsbHLH067/068/069-LAX1 module is essential for the regulation of inflorescence AM development in rice. With the onset of inflorescence AM development, the boundary expressed *LAX1* might be required for auxin signal transduction to promote inflorescence AM initiation [20]; at the inflorescence AM development stage, OsbHLH067/068/069 interact with LAX1 and mainly participate in metabolism pathways, fine-tuning the hormone and nutritional signaling to maintain inflorescence AM development. In addition, both OsbHLH067/068/069 and LAX1 are bHLH transcription factors, and thus their self-regulation of molecular networks in meristem development requires further examination.

## Materials and methods

### Plant materials and growth conditions

The *nsp1-D* (03Z11CH32) and *lax1* mutants (03Z11JS33) were identified from our T-DNA insertion mutant library [32]. All rice plants used in the study were derived from *Oryza sativa Japonica* variety Zhonghua 11 (ZH11), which was designed as wild type (WT). Rice plants were cultivated during the normal growing season in the experimental field of Huazhong Agricultural University in Wuhan, China (latitude 30.5˚N, 15m above sea level; average daily temperature approximately 28˚C).

### Isolation of the flanking sequences of T-DNA in *nsp1-D*

Thermal asymmetric interleaved polymerase chain reaction (PCR) was adopted to isolate the flanking sequence of T-DNA in *nsp1-D* [33]. In the 3-step PCR reactions, one end primer was on the T-DNA left border, namely IN-R, TL14, and LBT2 in turn, and the other end primer was always random primer AD8. The 3rd-round PCR product was sequenced and aligned.

### Plasmid construction for generation of *OsbHLH069* overexpressing plants

The *Cauliflower mosaic virus* (CaMV) 35S enhancer sequence was amplified from *pCAMBIA2301* (*pC2301*) and cloned to *pC2301* to construct the *35S-pC2301* vector. To construct the *35S-pOsbHLH069::OsbHLH069* vector, two fragments were obtained. One fragment was an approximately 3 kb promoter region of *OsbHLH069* amplified from the ZH11 genome using the primer pair 35S-pC2301-OsbHLH069-L1/35S-pC2301-OsbHLH069-R1. Another fragment was composed of approximately 2 kb of the *OsbHLH069* genome and a 1 kb region behind the termination codon amplified from the ZH11 genome using the primer pair 35S-pC2301-OsbHLH069-L2/35S-pC2301-OsbHLH069-R2. Using Gibson Assembly Master Mix (E2611S: New England BioLabs, Ipswich, MA, USA), the two fragments were cloned to the linearized vector *35S-pC2301* digested by *Eco*RI. The constructed plasmid was introduced into *Agrobacterium tumefaciens* EHA105, and finally transformed into rice callus to generate transgenic plants [32].

### Generation of CRISPR lines

*OsbHLH067*, *OsbHLH068*, and *OsbHLH069* were mutated using the CRISPR/Cas9 technique [63]. Two gRNA targets for each gene were selected. Two sgRNA sites shared by *OsbHLH067* and *OsbHLH069* were chosen to generate the double mutant *Osbhlh067 Osbhlh069*. Homozygotes of *Osbhlh068* and *Osbhlh069* were identified to develop the double mutant *Osbhlh068 Osbhlh069*. The CRISPR/Cas9 vector of *OsbHLH068* was transformed into *Osbhlh067* callus to obtain the double mutant *Osbhlh067 Osbhlh068* and into *Osbhlh067 Osbhlh069* callus to develop the triple mutant *Osbhlh067 Osbhlh068 Osbhlh069*. These resultant transgenic plants were identified by 2.5% agarose gel electrophoresis and sequencing.

### Scanning electron microscopy (SEM) analysis

For scanning electron microscopy (SEM), young panicles from wild type and *nsp1-D* mutants at typical development period were carefully dissected to remove bract hairs and keeping tissue integrity. The samples were fixed in 2.5% glutaraldehyde (2.5% GA in a 50mM phosphate buffer, pH 7.0) at 4˚C overnight, dehydrated with an ethanol series of from 25% to 100%, and dried. Then, the tissues were coated by using an E-100 ion sputter, and observed under a scanning electron microscope (S570, Hitachi, Tokyo, Japan).

## Histology and *in situ* hybridization

Young panicles from wild type and *nsp1-D* as well as *Osbhlh067 Osbhlh068 Osbhlh069* triple mutant at different developmental stages were fixed with 50% FAA solution containing 50% ethanol, 3.7% formaldehyde, and 5% acetic acid at 4˚C overnight. The samples were dehydrated by gradient ethanol and made transparent by xylene, followed by embedding in paraffin, and then sliced into 8 μm sections for 0.5% toluidine blue staining and *in situ* hybridization.

The templates for the *OsbHLH067*, *OsbHLH068*, *OsbHLH069*, *LAX1*, and *OSH1* probes were amplified from ZH11 cDNA using gene-specific primers joined with T7 or SP6 promoters as previously reported [59]. Probes were synthesized using a digoxigenin (DIG)-labeling kit (Millipore Sigma, Burlington, MA, USA). Hybridizations were conducted at 50˚C overnight for total probes as previously described [64].

## Yeast two-hybrid assay

The Matchmaker Gold yeast two-hybrid system (Clontech Laboratories, Mountain View, CA, USA) was used. The CDS of *OsbHLH067*, *OsbHLH068*, *OsbHLH069*, and *LAX1* was amplified from ZH11 cDNA and then cloned to pGBK-T7 and pGAD-T7, respectively. The pGAD-OsbHLH003 was used as a negative control [65]. Truncated fragments of LAX1 (1–159) and LAX1 (31–99) were obtained by PCR amplification and then cloned to pGBK-T7. Combined constructs were transformed into AH109 strains in an AD-BK-coupled manner.

## Subcellular localization and bimolecular fluorescence complementation (BiFC) assay

To test the subcellular localization of OsbHLH067, OsbHLH068 and OsbHLH069, their coding sequences (CDS) without the termination codon were separately amplified from ZH11 cDNA and then ligated in-frame to PM999-GFP containing a CaMV35S promoter at the N-terminal end and a GFP coding sequence at the C-terminal end. To obtain the BiFC constructs, the CDS of *OsbHLH067*, *OsbHLH068*, *OsbHLH069*, and *LAX1* was amplified and cloned into pSCYNE (nCFP) and pSCYCE (cCFP) [66]. The tested constructs were transformed into rice protoplasts as previously reported [67]. After incubation overnight at 23˚C, fluorescence in the transformants was observed using the Olympus FV1000.

## *In vitro* pull-down assays

To examine the interaction between OsbHLH069 and LAX1, the coding sequences of OsbHLH069 and LAX1 were separately cloned into the pGEX-4T-1 and pMAL-c2X vector to generate GST-OsbHLH069 and MBP-LAX1, respectively. For the interaction between OsbHLH067/OsbHLH068 and LAX1, the His-fused proteins of His-OsbHLH067 and His-OsbHLH068 were produced in the pET-32a vector.

For *in vitro* LAX1 and OsbHLH067/068/069 interaction, bacterial lysates containing ~15 mg MBP-LAX1 fusion protein was mixed with lysates containing ~30 mg GST-OsbHLH069 or His-OsbHLH067 or His-OsbHLH068 fusion proteins, respectively. Glutathione sepharose (30 μL; GE Life Sciences) was added to MBP-LAX1 and GST-OsbHLH069 combined solution or Ni Sepharose (30 μL; GE Life Sciences) to MBP-LAX1 and His-OsbHLH067/068 combined solution with rocking at 4˚C for 60 min. Beads were washed four times with the TGH buffer (50 mM HEPES, pH 7.5, 1.5 mM MgCl2, 150 mM NaCl, 1 mM EGTA, pH 8.0, 1% Triton X-100, 10% glycerol, 1 mM PMSF, and 1x Complete protease inhibitor cocktail [Roche]), and the isolated proteins were further separated on a 12% SDS-PAGE gels.

Then, the isolated proteins were detected by immunoblot analysis with anti-GST antibody (Abmart) and anti-MBP antibody (NEB) for LAX1-OsbHLH069 interaction, respectively. For LAX1-OsbHLH067 and LAX1-OsbHLH068 interactions, the isolated proteins were detected by immunoblot analysis with anti-His antibody (Abmart) and anti-MBP antibody (NEB), respectively. The immunoblot bands were visualized on a chemiluminescent imaging system (Tanon-5200, Tanon Science and Technology).

## RNA extraction and quantitative RT-PCR

Total RNA was extracted from various tissues of plants by using the TRIzol reagent (Invitrogen) according to the manufacturer's instructions, and first-strand cDNA was synthesized from 2.5 μg of total RNA with SuperScript III Reverse Transcriptase (Invitrogen) and oligo (dT)18 primer (Takara). By using the Applied Biosystems 7500 real-Time PCR system, qRT-PCR experiments were performed with SYBR Green Master Mix (Roche) in a total 10 μL reaction system according to the manufacturer's instructions, and the resultant data were normalized by the internal rice ubiquitin (UBQ) gene and analyzed by using the relative quantification method (2(-Delta Delta CT)). S5 Table lists the primers used in the qRT-PCR assays.

## RNA-seq analysis

Total mRNA from young panicles (<2 mm) of wild type and the *Osbhlh067 Osbhlh068 Osbhlh069* triple mutant with three biological replicates was isolated using Tri Reagent (Sigma-Aldrich, St. Louis, MO, USA) and purified using oligo(dT)-attached magnetic beads. The quality was checked using the NanoDrop 8000 Spectrophotometer (Thermo Fisher Scientific, Waltham, MA, USA) and Bioanalyzer 2100 (Agilent Technologies, Santa Clara, CA, USA). The mRNA libraries were constructed and sequenced using the DNBSeq platform at BGI Genomics (Shenzhen, PRC). High-quality reads were filtered with SOAPnuke (version 1.5.2) and then mapped to the rice reference genome IRGSP-1.0 using HISAT2 (version 2.0.4). Gene expression level was calculated with RSEM (version 1.2.12) after alignment with Bowtie2 (version 2.2.5). Differential expression analysis was conducted using the DESeq2 (version 1.4.5). Genes whose expression had a Q value ≤ 0.05 and fold change > 1.5 were chosen for further analysis. A Kyoto Encyclopedia of Genes and Genomes (KEGG) enrichment analysis of annotated DEGs was conducted by Phyper (https://en.wikipedia.org/wiki/Hypergeometric_distribution) based on the Hypergeometric test.

## Statistical analysis

The two-tailed Student's *t*-test in Microsoft Excel (Redmond WA, USA) was used for comparing means between two samples. A Dunnett's test was used for multiple comparisons by one-way analysis of variance (ANOVA) in the IBM SPSS Statistics software application (version 25.0: IBM, Armonk, NY, USA).

## Primers

All primers used in this study are listed in S5 Table.

## Accession numbers

Sequence data in this study can be found in the Rice Genome Annotation Project (http://rice.uga.edu/) under the following accession numbers: *NSP1* (*OsbHLH069*, LOC_Os01g57580), *LAX1* (LOC_Os01g61480), *OSH1* (LOC_Os03g51690), *OsbHLH065* (LOC_Os04g41570),

*OsbHLH066* (LOC_Os03g55220), *OsbHLH067* (LOC_Os05g42180), *OsbHLH068* (LOC_Os04g53990), and *OsbHLH070* (LOC_Os08g08160).

## Supporting information

**S1 Fig. Phylogenetic analysis of OsbHLH069.** (A) Phylogenetic analysis of putative OsbHLH069 homologs in rice using MEGA5.1 with neighbor-joining method and the following parameters: Poisson correction, pairwise deletion, and bootstrap (1,000 replicates; random seed). (B) Alignment of OsbHLH067, OsbHLH068, OsbHLH069, and OsbHLH070. Alignment was conducted with ClustalW from a MEGA5.1 program and then mapped using Gene-DOC software. Red, orange, and yellow shading represent residues conserved in 100%, 80%, and 60% of the sequences, respectively. (C) Comparison of the bHLH domain of OsbHLH067, OsbHLH068, OsbHLH069, and OsbHLH070.
(TIF)

**S2 Fig. Expression profiles and subcellular localization of OsbHLH067/068/069.** (A) to (C) Expression levels of *OsbHLH069* (A), *OsbHLH067* (B), and *OsbHLH068* (C) in various organs, including R (root), C (culm), L (leaf), LS (leaf sheath), and P (panicle < 5 mm). Rice *UBQ* gene acted as a control. Values represent means ± SEM from nine replicates. (D) Subcellular localization of OsbHLH069, OsbHLH067, and OsbHLH068 in rice protoplasts. The LAX1-RFP vector was used as a nuclear marker. The 35S-GFP vector served as a control. Bars = 10 μm.
(TIF)

**S3 Fig. Mutation site analysis of single, double, and triple mutants of *OsbHLH067, OsbHLH068*, and *OsbHLH069*.** (A) Schematic diagram of sgRNA sites of *OsbHLH069*, *OsbHLH067*, and *OsbHLH068* by CRISPR/Cas9 system, respectively. Boxes denote exons, and lines between the boxes indicate introns. (B) to (H) Analysis of the mutation sites in single mutants *Osbhlh069* (B), *Osbhlh067* (C), *Osbhlh068* (D), double mutants of *Osbhlh067 Osbhlh069* (E), *Osbhlh068 Osbhlh069* (F), *Osbhlh067 Osbhlh068* (G), and triple mutant of *Osbhlh067 Osbhlh068 Osbhlh069* (H). The red ellipsis represents the missing base; the double slash represents the omitted base; the bases underlined correspond to the target sequences; the bases in blue represent PAM.
(TIF)

**S4 Fig. Characterization of single mutant of *Osbhlh067, Osbhlh068*, and *Osbhlh069*.** (A) and (B) Comparison of the gross plant (A) and panicle (B) among WT, *Osbhlh067*, *Osbhlh068*, and *Osbhlh069* during reproductive growth. Bars in (A) and (B) = 20 cm and 4 cm, respectively. (C) to (H) Quantification of the number of tillers (C), PBs (D), SBs (E), SPBs (F), SSBs (G), and total spikelets (H) among WT, *Osbhlh067*, *Osbhlh068*, and *Osbhlh069*. Values in (C) to (H) are shown as means ± SEM from 12 replicates. Different letters denote significant differences ranked by the Dunnett's test (one-way analysis of variance, P < 0.05).
(TIF)

**S5 Fig. Expression relationship between *OsbHLH067/068/069* and *LAX1*.** (A) and (B) Gross morphology of WT (A) and *lax1* (B) during reproductive growth. Bars = 20 cm. (C) to (E) Panicle morphology of WT (C), *lax1/+* (D), and *lax1* (E). Bars = 4 cm. (F) The T-DNA insertion site in *lax1*. Box represents the *LAX1* genome, and the triangle indicates T-DNA. The primers L1, R1, and URB4 used for genotype analysis are marked with arrows. (G) The co-segregation analysis of *lax1*. W, H, and M indicate WT, heterozygous, and homozygous for T-DNA insertion, respectively. (H) Expression analysis of *OsbHLH069*, *OsbHLH067*,

*OsbHLH068*, and *LAX1* in the young panicles (< 2 mm) of *lax1*. (I) Quantitative RT-PCR analysis of *LAX1* expression in the young panicles (< 2 mm) of WT and triple mutant *Osbhlh067 Osbhlh068 Osbhlh069*. (J) to (L) *In situ* hybridization with a *LAX1* probe on WT (J) and *Osbhlh067 Osbhlh068 Osbhlh069* triple mutant (K, L) inflorescences at the differentiation stage of secondary branch meristem. Bars = 100 μm. The rice *UBQ* gene was used to normalize gene expression. The values shown in (H) and (I) are means ± SEM from nine replicates. Significant difference (two-tailed Student's *t*-test, ***$P < 0.001$); ns indicates not significant (two-tailed Student's *t*-test; $P > 0.05$).
(TIF)

**S6 Fig.** Quantitative RT-PCR analysis of eight auxin-related genes in the young panicles (< 2 mm) of *lax1* (A) and *Osbhlh067 Osbhlh068 Osbhlh069* triple mutants (B). Data were normalized to the rice UBQ gene, and values represent means ± SEM from three replicates. Significant difference (two-tailed Student's t-test, **$P < 0.01$, ***$P < 0.001$); ns indicates not significant (two-tailed Student's t-test; $P > 0.05$).
(TIF)

**S1 Table. Differentially expressed genes in the *Osbhlh067 Osbhlh068 Osbhlh069* triple mutant.**
(XLSX)

**S2 Table. KEGG terms from the down-regulated genes in the *Osbhlh067 Osbhlh068 Osbhlh069* triple mutant.**
(XLSX)

**S3 Table. KEGG terms from the up-regulated genes in the *Osbhlh067 Osbhlh068 Osbhlh069* triple mutant.**
(XLSX)

**S4 Table. Down-regulated genes involved in various biological processes in the *Osbhlh067 Osbhlh068 Osbhlh069* triple mutant.**
(XLSX)

**S5 Table. Primers used in the study.**
(XLSX)

## Acknowledgments

We thank Prof. Fang Yang for helpful comments.

## Author Contributions

**Conceptualization:** Tingting Xu, Changyin Wu.

**Funding acquisition:** Changyin Wu.

**Investigation:** Tingting Xu, Debao Fu, Xiaohu Xiong, Junkai Zhu, Zhiyun Feng, Xiaobin Liu.

**Methodology:** Tingting Xu, Debao Fu, Xiaohu Xiong.

**Supervision:** Changyin Wu.

**Visualization:** Tingting Xu, Debao Fu, Xiaohu Xiong, Changyin Wu.

**Writing – original draft:** Changyin Wu.

**Writing – review & editing:** Tingting Xu, Changyin Wu.

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
