## [Decision Letter · Decision Letter 0]

4 Aug 2022

Dear Dr Wu,

Thank you very much for submitting your Research Article entitled 'OsbHLH067, OsbHLH068, and OsbHLH069 act redundantly to regulate the formation of inflorescence  axillary meristems  in rice' to PLOS Genetics.

The manuscript was fully evaluated at the editorial level and by two independent peer reviewers. The reviewers appreciated the attention to an important problem, but raised some substantial concerns about the current manuscript. Based on the reviews, we will not be able to accept this version of the manuscript, but we would be willing to review a much-revised version. We cannot, of course, promise publication at that time.

If you decide to revise the manuscript for further consideration at PLOS Genetics, please aim to resubmit within the next 60 days, unless it will take extra time to address the concerns of the reviewers, in which case we would appreciate an expected resubmission date by email to plosgenetics@plos.org.

[LINK]

We are sorry that we cannot be more positive about your manuscript at this stage. Please do not hesitate to contact us if you have any concerns or questions.

Yours sincerely,

Yuling Jiao

Guest Editor

PLOS Genetics

Li-Jia Qu

Section Editor: Plant Genetics

PLOS Genetics

1. The authors are advised to reconsider the interpretation of the phenotype, as suggested by Reviewer 2.

2. The presentation of the work need to be substantially improved, including consistency, and also language.

Reviewer's Responses to Questions

**Comments to the Authors:**

Reviewer #1: General remarks:

1. Overall the language has to be improved. I see the problem not being native speaker (this applies to me too). Still somebody should try to correct the language.

2. The manuscript contains to many inconsistencies and errors. Material and methods are incomplete etc.

I mention some of the problems at the end of the review, but not all. It is not the job of the reviewer to indicate all errors.

Summary of the paper:

The authors provide valuable information on the following points. Most of the information provided may be helpful for other researchers in this field:

- The authors describe a novel rice mutant, nsp1-D, with reduced inflorescence branching. The mutant phenotype is semi-dominant.

- They give a detailed description of lateral meristem defects in heterozygous and homozygous mutant plants.

- This mutation is caused by the insertion of a T-DNA upstream of rice bHLH069 gene, causing over-expression of the gene. Insertion of the T-DNA co-segregated with the phenotypes and overexpressing bHLH069 by T-DNA-constructs led to reduction of inflorescence branching in varying amounts.

- bHLH069 is a member of a small group of related genes, including bHLH067 and bHLH68. qRT-PCR indicated, that the three genes are widely expressed, but in situ hybridization demonstrated preferential expression in shoot, inflorescence and floral meristems. Expression patterns differ from the expression pattern of the branching regulator LAX1.

- Nuclear localization has been demonstrated for GFP fusion of all three bHLH proteins.

- Loss of function (LOF) alleles have been obtained for the three genes by CRISPR/Cas9 technology.

- Single LOF lines did not display phenotypic alterations. Similarly, double mutants had no or very weak defects in branching.

- Triple mutant plants displayed reduced height, as well as reduced shoot and inflorescence branching. Reduced inflorescence branching is linked to reduced expression of the meristem marker gene OSH1.

- The authors identify a new LAX1 loss of function allele in the isogenic background. The phenotype of this mutant is comparable to the phenotypes described for lax1-mutants described earlier.

- Combining lax-1 with nsp-1D resulted in enhanced lateral meristem defects. Heterozygous nsp1-D in lax-1 mutant background and heterozygous lax-1 in homozygous nsp1-D mutant background enhanced the phenotypic defects of the respective mutants. (statistical analysis seems to assume normal distribution of data – but at least for F,G,H, and I data are shifted towards 0).

- In situ-hybridization experiments demonstrated normal LAX-1 expression in the bHLH triple mutant, qRT may have indicated slightly enhanced LAX-1 expression I the triple bHLH line (statistical analysis missing).

- Expression level of bHLH067,068 and 069 is not affected by the lax-1 mutation.

- In yeast two-hybrid assays truncated versions of bHLH067/8/9 physically interacted with a truncated form of LAX1. The same interactions were found when only the bHLH domains of the respective genes were used.

- In bimolecular fluorescence complementation (BiFC) assays the interactions of of the bHLH-fragments could be confirmed.

- RNASeq experiments using material from wild-type and triple bHLH-mutants resulted in a high number of differentially expressed genes.

- Among the genes up- or down-regulated in the triple-mutant, several functional categories are enriched. One category was significantly enriched in the genes down-regulated (Protein processing in endoplasmic reticulum; Q value=9.77E-04), several categories were enriched in the genes up-regulated in the triple mutant.

Do the experiments confirm statements of the authors?

1. A weak point of the manuscript is the fact, that there is no explanation for the contradicting branching defects:

Over-expression of bHLH069 leads to reduced lateral meristem formation in the inflorescence. This indicates a repressing function of bHLH069 in meristem branching.

A combination of loss-of-function alleles for bHLH067/068/069 leads to reduced inflorescence branching. This indicates an activating function of bHLH069 in meristem branching.

This question is not addressed in the paper.

2. The authors need to give a functional explanation or at least a hypothesis for the results of the different experiments testing the interaction of bHLH067/68/69 and LAX1:

Double mutant analysis shows mutual enhancement of mutant phenotypes, which is most easily explained, when LAX1 and the three bHLH genes act in independent pathways/in parallel. Y2H and BiFC show a physical interaction of partners – the partners act in a protein complex. Either both partners are needed to have an active complex or one of the partners inactivates the other when forming a complex. In the first case, removing one of the partners should have the same effect as removing both (no enhancement of phenotypic defects). In the second case, effects of mutations should be antagonistically more branching in one of the mutants, less branching in the other).

This question needs to be addressed.

3. The authors try to clarify the function of bHLH067/68/69 by analyzing transcript accumulation in the triple bHLH mutant. They find functional categories of genes enriched in up- or down-regulated genes. Why did they select a category with a relatively poor Q value for further analysis and not the one with the highest Q-value?

As the authors seem to be interested to define direct targets of bHLH067/68/69, they should try to clarify, if these bHLH proteins are transcription activators or repressors. Based on the description of the Y2H experiments they are acting as transcription activators. This would indicate that the category “Protein processing in endoplasmic reticulum”, down-regulated in the triple-mutant would be the best candidate for further analysis.

The authors need to address this question.

Minor concerns:

The authors should try to explain the differences/contradictions between qRT-PCR experiments and RNA in-situ hybridization.

The authors need a different negative control for Y2H and biFC. bHLH proteins have a tendency to form dimers. Binding of proteins is a chemical process. Using unnatural high concentrations of proteins that would not interact in natural concentrations may indicate interactions that are not relevant under

As bHLH-genes are numerous, it is necessary to prove that LAX binds better to bHLH067/68/69 than to other bHLH proteins not linked to branching. This would mean, that the negative control for Y2H and biFC would not be an empty vector control.

The explanations, why phenotypes of genes with sequence similarity to bHLH067/68/69 in Arabidopsis thaliana have not been linked to similar phenotypes (branching vs. salt stress response and flowering transition; page 16) are unclear.

RNA-Seq experiments: Information is insufficient: Materials and Methods does not describe the analysis in sufficient detail to jugde the results. Why are only 6740 genes included in the DeSeq-analysis? What filters have been used to obtain the number 6740 genes?

Page 30 Accession numbers: These are gene names, but not accession numbers. I tried to download the corresponding sequences and found this difficult. As gene names change, accession numbers must be used.

References to supplemental tables S2 and S3 need to be switched in in lines 290 and 292.

Description of qRT is missing in Materials and Methods. In addition it has been demonstrated, that using a single reference gene can give wrong results, especially when comparing expression levels between different tissues.

Etc.

Reviewer #2: Please check the attachment

**Have all data underlying the figures and results presented in the manuscript been provided?**

Reviewer #1: Yes

Reviewer #2: None

PLOS authors have the option to publish the peer review history of their article (what does this mean?). If published, this will include your full peer review and any attached files.

Reviewer #1: No

Reviewer #2: No

---

## [Decision Letter · Decision Letter 1]

17 Nov 2022

Dear Dr Wu,

Thank you very much for submitting your Research Article entitled 'OsbHLH067, OsbHLH068, and OsbHLH069 act redundantly to regulate the formation of inflorescence axillary meristems in rice' to PLOS Genetics.

The manuscript was fully evaluated at the editorial level and by two independent peer reviewers. Note that Reviewer 1 is no longer available so that a new reviewer has been involve, who has taken the previous reviewers' comments into account. Both reviewers appreciated the attention to an important problem as well your effects in the first round of revision. However, they both raised some substantial concerns about the current manuscript. Based on the reviews, we will not be able to accept this version of the manuscript, but we would be willing to review a much-revised version. We cannot, of course, promise publication at that time.

If you decide to revise the manuscript for further consideration at PLOS Genetics, please aim to resubmit within the next 60 days, unless it will take extra time to address the concerns of the reviewers, in which case we would appreciate an expected resubmission date by email to plosgenetics@plos.org.

We are sorry that we cannot be more positive about your manuscript at this stage. Please do not hesitate to contact us if you have any concerns or questions.

Yours sincerely,

Yuling Jiao

Guest Editor

PLOS Genetics

Li-Jia Qu

Section Editor

PLOS Genetics

1. Please clarify the difference between mutant and overexpression lines, in both of which branch meristem formation is compromised. Given the difference in phenotype, these two scenarios are likely caused by different mechanisms.

2. The writing needs to be substantially polished.

3. Please make sure all reviewers' comments, including those from the first round, are fully addressed.

Reviewer's Responses to Questions

**Comments to the Authors:**

Reviewer #2: The revised version by Xu et al., addressed some points, but I think there are still some concerns are not well addressed.

1. As shown in the manuscript, OsbHLH69/68/67 are important for rice panicle development, and both over-expression and loss of function mutants are compromised in panicle development. During rice panicle development, primary and secondary branching meristems still have the meristematic ability, but spikelet meristems are somehow developed into a terminate status. So it is very crucial to clearly clarify the defects of these mutants. As I suggested last time, OSH1 is not a good marker here, it is better to use some more specific marker, TAWAWA1 as branch meristem maker, and FZP and/or LHS1(OsMADS1) as spikelet meristem specific marker. And I would also suggest the authors discuss potential roles of OsbHLH69/68/67 in meristem determinacy or indeterminacy regulation.

2. In figure 8, selected AM formation genes are not enough, and their functions are quite different. As I suggested, it is better to include nsp1-D mutant inside and check some key transition MADS-box genes.

Minor concerns

1. Line 31, I would suggest to use a weaker statement, it is not convinced that the phenotypes of lax1 and nsp1-D are "similar".

2. For BiFC part, empty nCFP or cCFP are not suitable negative controls.

Reviewer #3: The research article by Xu et al. described the isolation and cloning of a T-DNA insertion mutant with sparse-panicle phenotype. Besides identifying the causal gene locus, they characterized the mutant phenotype by morphological description and ISH and confirmed through molecular genetic approaches that the sparse-panicle phenotype was indeed induced by overexpression of a gene encoding a bHLH transcription factor. They further profiled the spatio-temporal expression patterns of this bHLH069 gene and its close paralogs bHLH067 and bHLH068, which likely function redundantly with bHLH069 in promoting AM initiation. They also found that the bhlh067 bhlh068 bhlh069 triple mutant resembles Oslax1 mutant in AM initiation potential, suggesting that these bHLH TFs might function in the same pathway with OsLAX1. They performed a number of in vitro assays to confirm that OsbHLH 067/068/069 interact with LAX1. They carried out high-throughput transcriptome profiling and found some potential targets of the common module composed of OsbHLH 067/068/069-OsLAX1.

Overall, this study is sound and interesting, not only to the field of crop sciences, but also to the field of plant development. It is definitely a beneficial complement to currently published studies. However, the underlying mechanism of the OsbHLH 067/068/069-OsLAX1 module is still lacking. There are still some major issues and quite some minor problems for the authors to address.

Major concerns

1. The authors need to provide a reasonable explanation for why both the nsp-1D, representing an overexpression line of OsbHLH069, and the osbhlh067 068 069 triple mutant show defective AM initiation. The authors should focus on the differences rather than the common behavior between the two types of mutants. Similar to the phenotypic analysis for nsp-1D, SEM or histological observations against loss-of-function mutants should be included to address this.

2. Is it possible that the triple mutant and the oslax1 mutant are defective in AM initiation mainly due to nutritional defects, which cause pleiotropic phenotypes including AM initiation defects, while in the overexpression lines the defect is caused by disequilibrium in OsbHLH-OxLAX1 complex?

3. From the phylogenetic tree in figure S1, it can be inferred that OsbHLH068 is more similar to OsbHLH070, as compared to OsbHLH067 and OsbHLH069. It seems the authors had pre-excluded OsbHLH070 in the AM-regulating module for some reason, but failed to provid the reason in the current version. The authors need to fill this gap to show why and how they pre-excluded OsbHLH070 in their analysis. To this end, the in situ hybridization analysis against OsbHLH070 may be required.

4. Considering that the expression region of OsbHLH067/068/069 mRNA is much wider than that of LAX1, the loss of function mutants of the bHLH genes would lead to defects other than that seen in lax1 mutant. The authors need to reconcile the discrepancy. To answer this question, it may be necessary to create translational reporters, i.e., fused proteins of OxbHLH/OsLAX1 and fluorescent protein tags, to show that the protein products coexist in leaf axils or rising meristems.

5. The genetic interactions between OsbHLH067/068/069 and OsLAX1 should be characterized by generating the quadruple mutant containing Osbhlh067 068 069 and oslax1 loss of function mutations and analyzing the corresponding phentoypes. This is also the experiment suggested by Reviewer 1.

6. To conclude that the downregulated genes are actually downstream of OsbHLH067/068/069, it is necessary to perform transcriptome analysis also in nsp1-D, or at least qRT-PCR analysis is necessary to examine the expression of the downregulated genes in nsp1-D and/or the generated bHLH067/068/069 overexpression lines.

Minor concerns

1. The language needs to be improved substantially.

2. Materials and methods need to be elaborated on. Please provide detailed information instead of referring readers to some other papers.

3. The authors should use two or more reference genes in qRT analysis to avoid the bias induced by using only a single reference gene, the expression level might itself differ among tissues.

4. Line edits (should not limited to these listed ones)

L30: delete “which”

L31: LAX1 -> which, transcriptomic -> transcriptomic

L32: metabolism -> metabolic

L41: has -> have

L44: underling -> underlying

L89: homologues -> homologous

L90: mutation -> mutations； auxin pathways -> auxin signaling pathways

L92: OsPID is not an auxin transporter, but a kinase activating PIN1.

L96: delete “rice”, which appears twice in the same sentence； Factor -> factors

L137: performed -> exhibited the phenotype of weakly sparse inflorescences (same for L176)

L141: effects -> defects

L173: genotype -> insertion

L175: seriously -> severely

L232: statue -> stature

L253: transcription -> transcriptional

L259: less -> few

L289: performed -> perform

L301: factor -> factors

L314: functions -> function; interacts ->interact

L341: an obviously defects AM development -> an obvious defect in AM development

L350: In analyzing -> By analyzing

L357: delete “the”

L429: Isolation flanking -> isolation of the flanking

**Have all data underlying the figures and results presented in the manuscript been provided?**

Reviewer #2: None

Reviewer #3: Yes

PLOS authors have the option to publish the peer review history of their article (what does this mean?). If published, this will include your full peer review and any attached files.

Reviewer #2: No

Reviewer #3: No

---

## [Decision Letter · Decision Letter 2]

24 Jan 2023

Dear Dr Wu,

Thank you very much for submitting your Research Article entitled 'OsbHLH067, OsbHLH068, and OsbHLH069 redundantly regulate inflorescence axillary meristem formation in rice' to PLOS Genetics.

The manuscript was fully evaluated at the editorial level and by two independent peer reviewers. The reviewers appreciated the attention to an important topic but were unsatisfied with your revision. In fact, both reviewers have asked you to better address concern they identified in the previous rounds of revisions, which we ask you better address in a revised manuscript.

We therefore ask you to modify the manuscript according to the review recommendations. Your revisions should address the specific points made by each reviewer.

Yours sincerely,

Yuling Jiao

Guest Editor

PLOS Genetics

Li-Jia Qu

Section Editor

PLOS Genetics

Both reviewers found that the majority of their comments were unaddressed, which make both of them unsatisfied with the revision. There are two types of concerns. The first group of concerns is about the principles of axillary meristem development. As mentioned by Reviewer 2, vegetative and reproductive stage axillary meristem are regulated by distinct groups of genes. Also, branch meristems and spikelet meristems are different. The second group of concerns is about clarification of the OsbHLH genes function. Reviewer 3 suggests clarifying the difference between overexpressor and mutant phenotypes, and using qRT-PCR analysis to verify the expression of the downregulated genes in nsp1-D and other mutants. I would like to ask the authors to at least address the above mentioned points.

A minor point:

Arabidopsis auxin influx carrier LAX1 mentioned in the Introduction is not relevant to the paper, and more importantly confusing with rice LAX1 that regulates spike branching.

Reviewer's Responses to Questions

**Comments to the Authors:**

Reviewer #2: Xu et al., addressed some points in the revised version, but I think there are still some points should be discussed further.

1. Vegetative axillary meristems (AMs) and reproductive AMs development are quite different in rice. Both bHLH69 over-expression lines and bHLH67/68/69 triple mutants exhibited branching defects, while triple mutants are compromised in vegetative and reproductive stages. I think it is important to discuss in more details. And I suggest the authors to clearly clarify branching defects caused by AM initiation or outgrowth, eg, discussion part from line 406, it is rather confusing if AM initiation or outgrowth are not discussed separately.

2. In rice, identities of branching meristems and spikelet meristem are different. As shown here that branching meristems seems normally formed in *nsp1-D* mutants, but could not develop into spikelet meristem. The determinacy balance between inflorescence meristem and spikelet meristem are very important for final panicle structure. It is essential to discuss the potential role of newly identified bHLH69 and its homologs in regulating the balance between inflorescence termination and spikelet meristem formation.

3. bHLH67/68/69 are widely expressed in almost whole meristem, and LAX1 is mainly expressed in the boundary region. Even though LAX1 protein could move, but it is still difficult to conclude real biological function of the interactions between LAX1 and bHLH69 related proteins. I think the interaction part somehow leads to more complicate to understand function of bHLH69 and its related homologs.

Reviewer #3: Overall, this study will be a beneficial supplement to the field. I can see that the language has been substantially improved and the authors have provided more details in Materials and Methods section.

However, the underlying mechanism of the OsbHLH 067/068/069-OsLAX1 module is still lacking. In addition, the concerns raised by me and other reviewers were not fully addressed. For nearly all the major concerns, the authors did not perform additional experiments to answer those questions, but threw them off very easily and promised casually to solve them in their future study. As I mentioned in last review, the authors need to reconcile the apparent discrepancy before the work can be accepted.

Major concerns

1. L201 The authors claimed that OsbHLH070 does not contain a bHLH domain, which is different from their response to reviews and the alignment shown in Fig S1B. Even given the bHLH domain in OsbHLH070 is longer than those in the other three paralogues, the functions of OsbHLH070 are not necessarily different from the other three. The authors should not have precluded this gene in their functional characterization.

2. The authors should perform in situ hybridization for OsbHLH067/068/069 in lax1 mutant to exclude the transcriptional regulation of LAX1 on OsbHLH067/068/069.

3. The authors should perform in situ hybridization for representative genes (such as those required for AM initiation and sucrose metabolism) identified from RNA-seq analysis in the background of Osbhlh triple mutant and lax1 mutant, especially because the authors claimed them to be downstream genes of OsbHLH067/068/069 and LAX1. qRT-PCR is not sufficient.

4. The authors should explain why both the nsp-1D, representing an overexpression line of OsbHLH069, and the osbhlh067 068 069 triple mutant show defective AM initiation. The authors should focus on the differences rather than the common behavior between the two types of mutants. To this end, they should focus on the indeterminacy of distinct types of meristems in different mutants. Checking the expression of more meristem markers by in situ hybridization such as TAWAWA1, FZP and LHS1 may help distinguish the subtle differences between them. Furthermore, similar to the phenotypic analysis for nsp-1D, at least SEM or histological observations against loss-of-function mutants should be included to address this.

5. Considering that the expression region of OsbHLH067/068/069 mRNA is much wider than that of LAX1, the loss of function mutants of the bHLH genes would lead to defects other than that seen in lax1 mutant. The simple OsbHLH067/068/069-LAX1 module is not sufficient. The authors should create translational reporters, to show that in planta the protein products coexist in leaf axils or rising meristems.

6. The genetic interactions between OsbHLH067/068/069 and OsLAX1 should be characterized by generating quadruple mutant containing Osbhlh067 068 069 and oslax1 loss of function mutations and analyzing the corresponding phentoypes. This is also the experiment suggested by Reviewer 1.

7. To conclude that the downregulated genes are actually downstream of OsbHLH067/068/069, it is necessary to perform at least qRT-PCR analysis is necessary to examine the expression of the downregulated genes in nsp1-D and/or the generated bHLH067/068/069 overexpression lines.

Minor concerns

1. A number of highlighted regions are identical to last version or not pinpointing at all. The authors should pinpoint what has been changed and to which review is the revision relevant.

2. The third paragraph in the Introduction (L86-98), which focuses on auxin biosynthesis, transport and signaling, does not seem to be relevant to the study. None of the genes or pathways were involved in the phenotypes.

3. L89 PID is different from PIN1 and AUX/LAX and not an auxin carrier, as pointed out in last review.

4. L337 obviously -> obvious, as pointed out in last review.

5. L338 observation ->observations

6. L529 Gutathione -> Glutathione

**Have all data underlying the figures and results presented in the manuscript been provided?**

Reviewer #2: None

Reviewer #3: Yes

PLOS authors have the option to publish the peer review history of their article (what does this mean?). If published, this will include your full peer review and any attached files.

Reviewer #2: No

Reviewer #3: No

---

## [Decision Letter · Decision Letter 3]

8 Mar 2023

Dear Dr Wu,

We are pleased to inform you that your manuscript entitled "OsbHLH067, OsbHLH068, and OsbHLH069 redundantly regulate inflorescence axillary meristem formation in rice" has been editorially accepted for publication in PLOS Genetics, pending on some minor revisions (please see comments from Reviewer 2). Congratulations!

Furthermore, before your submission can be formally accepted and sent to production you will need to complete our formatting changes, which you will receive in a follow up email. Please be aware that it may take several days for you to receive this email; during this time no action is required by you. Please note: the accept date on your published article will reflect the date of this provisional acceptance, but your manuscript will not be scheduled for publication until the required changes have been made.

Once your paper is formally accepted, an uncorrected proof of your manuscript will be published online ahead of the final version, unless you've already opted out via the online submission form. If, for any reason, you do not want an earlier version of your manuscript published online or are unsure if you have already indicated as such, please let the journal staff know immediately at plosgenetics@plos.org.

In the meantime, please log into Editorial Manager at https://www.editorialmanager.com/pgenetics/, click the "Update My Information" link at the top of the page, and update your user information to ensure an efficient production and billing process. Note that PLOS requires an ORCID iD for all corresponding authors. Therefore, please ensure that you have an ORCID iD and that it is validated in Editorial Manager. To do this, go to 'Update my Information' (in the upper left-hand corner of the main menu), and click on the Fetch/Validate link next to the ORCID field.  This will take you to the ORCID site and allow you to create a new iD or authenticate a pre-existing iD in Editorial Manager.

Yours sincerely,

Yuling Jiao

Guest Editor

PLOS Genetics

Li-Jia Qu

Section Editor

PLOS Genetics

Comments from the reviewers:

Both reviewers found the new version improved, but still feel previous comments were not sufficiently addressed. I would like to ask you to better address the reviewers' comments from the most recent round of review. Reviewer 2 has listed a few points that are most imminent.

Reviewer's Responses to Questions

**Comments to the Authors:**

Reviewer #2: Xu et al., addressed most of points in the revised version, I have some minor suggestions.

1. Authors put new real-time data about LAX1 function in auxin signalling and it seems not the case of bHLH69. It was somehow confusing that LAX1 and bHLH69 interact with other, but regulates different pathway. 

2. IM, BM, SM have different determinacy identities, I would suggest to discuss more about this issue. TAWAWA1 is a very important regulator in meristem phase transition, and the paper Yoshida et al., 2013 PNAS was ignored completely.

3. Line 90, "AUX" should be "AUX1"

4. Line 93, several genes are not "auxin signaling pathway", there are auxin biosynthesis genes.

5. Line 127, TB1 is not a "AM formation genes".

Reviewer #3: Although the concerns were not all addressed, it looks better.

**Have all data underlying the figures and results presented in the manuscript been provided?**

Reviewer #2: Yes

Reviewer #3: Yes

PLOS authors have the option to publish the peer review history of their article (what does this mean?). If published, this will include your full peer review and any attached files.

Reviewer #2: No

Reviewer #3: No

**Data Deposition**

http://datadryad.org/submit?journalID=pgenetics&manu=PGENETICS-D-22-00724R3

**Press Queries**

---

## [Editor Report · Acceptance letter]

4 Apr 2023

PGENETICS-D-22-00724R3 

OsbHLH067, OsbHLH068, and OsbHLH069 redundantly regulate inflorescence axillary meristem formation in rice 

Dear Dr Wu, 

We are pleased to inform you that your manuscript entitled "OsbHLH067, OsbHLH068, and OsbHLH069 redundantly regulate inflorescence axillary meristem formation in rice" has been formally accepted for publication in PLOS Genetics! Your manuscript is now with our production department and you will be notified of the publication date in due course.

With kind regards,

Timea Kemeri-Szekernyes

PLOS Genetics

On behalf of:
